# SPECTRAL BELLMAN METHOD:
# UNIFYING REPRESENTATION AND EXPLORATION IN RL

**Ofir Nabati**[1,4][*]**, Bo Dai**[2]**, Shie Mannor**[1,3]**, Guy Tennenholtz**[4]
[1] Technion, [2] Google DeepMind, [3] Nvidia Research, [4] Google Research

## ABSTRACT

Representation learning is critical to the empirical and theoretical success of reinforcement learning. However, many existing methods are induced from model-learning aspects, misaligning them with the RL task in hand. This work introduces the Spectral Bellman Method, a novel framework derived from the Inherent Bellman Error (IBE) condition. It aligns representation learning with the fundamental structure of Bellman updates across a *space* of possible value functions, making it directly suited for value-based RL. Our key insight is a fundamental spectral relationship: under the zero-IBE condition, the transformation of a *distribution* of value functions by the Bellman operator is intrinsically linked to the feature covariance structure. This connection yields a new, theoretically-grounded objective for learning state-action features that capture this Bellman-aligned covariance, requiring only a simple modification to existing algorithms. We demonstrate that our learned representations enable structured exploration by aligning feature covariance with Bellman dynamics, improving performance in hard-exploration and long-horizon tasks. Our framework naturally extends to multi-step Bellman operators, offering a principled path toward learning more powerful and structurally sound representations for value-based RL.

## 1 INTRODUCTION

Efficient reinforcement learning (RL) in complex environments hinges on two critical challenges: learning effective representations and performing efficient exploration. While many approaches tackle representation learning (Laskin et al., 2020; Schwarzer et al., 2021; Oh et al., 2015; Zhang et al., 2021; Nabati et al., 2023; Barreto et al., 2017; Zhang et al., 2022) and exploration as separate problems, a deeper synergy is needed, particularly for control tasks where features must support both accurate value estimation and strategic data gathering. This work introduces a novel framework to learn representations that inherently unify these aspects, paving the way for more powerful and sample-efficient RL agents.

Our approach is rooted in the theory of Inherent Bellman Error (IBE) (Zanette et al., 2020a). The IBE quantifies the suitability of a feature space for value-based RL by measuring the minimum error in representing Bellman updates. A zero IBE condition, generalizing Linear MDPs (Jin et al., 2020), is highly desirable as it implies the function space is closed under the Bellman operator. However, directly discovering features that satisfy this condition *a priori* remains a significant challenge through a min-max-min optimization (Modi et al., 2024), especially with general function approximator. The challenge impedes the practical applications of IBE-based representation learning.

This paper introduces the **Spectral Bellman Method (SBM)**, a framework which makes low-IBE representation learning tractable and addresses exploration upon the learned representation. Our approach uses a fundamental spectral relationship of IBE: when the IBE is zero, the transformation of a *distribution* of value functions by the Bellman operator is intrinsically linked to the covariance structure of the features themselves. This connection reveals that features aligned with Bellman updates naturally possess a covariance structure that can be exploited for structured exploration. Leveraging this insight, we derive a novel, theoretically grounded objective function. Our method learns state-action features whose covariance inherently captures this Bellman-aligned structure, while avoiding the complicated optimization. The learned representations not only enhance value

---

[*]ofirnabati@gmail.com

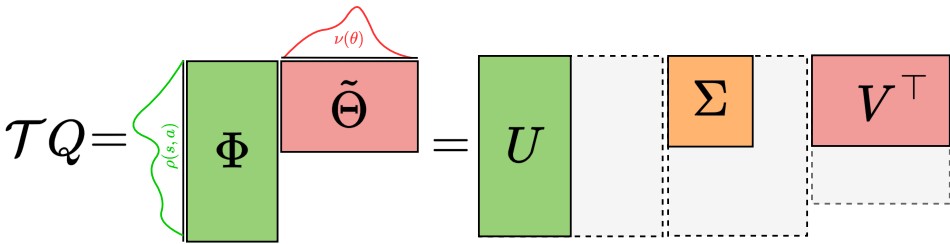

Figure 1: Spectral Representation of the optimal Bellman operator. Under zero IBE condition, the linear representation is equivalent to an SVD decomposition of rank $d$ with a singular value matrix $\Sigma$.

approximation but also naturally facilitate structured exploration strategies. Specifically, we use Thompson Sampling (TS) driven by the learned representation for efficient exploration.

Our contributions are as follows. **(1)** We propose a novel representation learning framework, the Spectral Bellman Method, which unifies representation learning with exploration, motivated by the structural implications of the Inherent Bellman Error. Our method requires only a simple modification to existing value-based RL algorithms. **(2)** We demonstrate the effectiveness of the Spectral Bellman Method and its learned representations on the Atari benchmark, including a subset of hard exploration games. **(3)** The extension of this representation learning approach to capture the structure of powerful multi-step Bellman operators and targets.

## 2    BACKGROUND: THE ZERO-IBE CONDITION AND EFFICIENT EXPLORATION

We consider a Markov Decision Process (MDP) (Puterman, 1990) $(\mathcal{S}, \mathcal{A}, \gamma, r, P)$ with state space $\mathcal{S}$, action space $\mathcal{A}$, discount $\gamma \in [0, 1)$, reward $r : \mathcal{S} \times \mathcal{A} \to \mathbb{R}$ $(|r(s, a)| \leq R_{\max})$, and transition kernel $P : \mathcal{S} \times \mathcal{A} \mapsto \Delta(\mathcal{S})$. The $Q$-function is $Q^\pi(s, a) = \mathbb{E}_{\pi, P}[\sum_{t=0}^{\infty} \gamma^t r(s_t, a_t) \mid s_0 = s, a_0 = a]$. The goal is to find an optimal policy $\pi^*(s) \in \arg\max_\pi \mathbb{E}_{a \sim \pi}[Q^\pi(s, a)]$.

The optimal Q-function $Q^*(s, a)$ satisfies $Q^*(s, a) = r(s, a) + \gamma \mathbb{E}_{s' \sim P(\cdot|s,a)}[\max_{a' \in \mathcal{A}} Q^*(s', a')]$. The optimal Bellman operator $\mathcal{T}$ on $Q : \mathcal{S} \times \mathcal{A} \to \mathbb{R}$ is defined by

$$\mathcal{T}Q(s, a) = r(s, a) + \gamma \mathbb{E}_{s' \sim P(\cdot|s,a)}\left[\max_{a' \in \mathcal{A}} Q(s', a')\right]. \qquad \text{(Optimal Bellman Operator)}$$

This is a special case of the policy Bellman operator $\mathcal{T}^\pi Q(s, a) = r(s, a) + \gamma \mathbb{E}_{s' \sim P(\cdot|s,a), a' \sim \pi(\cdot|s')}[Q(s', a')]$. Indeed, for a greedy policy $\pi$ w.r.t $Q$, $\mathcal{T}$ and $\mathcal{T}^\pi$ are equivalent.

### 2.1    THE ZERO INHERENT BELLMAN ERROR CONDITION

$Q^*$ is often approximated linearly with features $\phi : \mathcal{S} \times \mathcal{A} \to \mathbb{R}^d$ as $Q_\theta(s, a) = \phi(s, a)^\top \theta$ for $\theta \in \mathbb{R}^d$. We assume $\|\phi(s, a)\|_2 \leq 1$.

**Definition 1** (Function Space and Parameter Bounds)**.** *Let $\phi : \mathcal{S} \times \mathcal{A} \to \mathbb{R}^d$ be a feature map. We define the linear function space for Q-functions as:*

$$\mathcal{Q}_\phi = \{Q_\theta(s, a) = \phi(s, a)^\top \theta \mid \theta \in \mathcal{B}_\phi\}, \quad \mathcal{B}_\phi = \{\theta \in \mathbb{R}^d \mid \sup_{(s,a) \in \mathcal{S} \times \mathcal{A}} |\phi(s, a)^\top \theta| \leq D\},$$

*for a bounded parameter set $\mathcal{B}_\phi$, typically defined such that the resulting Q-values are bounded. for some maximum value $D$ (e.g., $D = R_{max}/(1 - \gamma)$).*

Effective exploration for policy learning and data collection is crucial in RL, especially with large state/action spaces, where good representations are key (Jin et al., 2020; Zanette et al., 2020a; Azizzadenesheli et al., 2018). While policy learning has advanced (Schulman et al., 2017; Kapturowski et al., 2018; Badia et al., 2020), representation for exploration remains challenging (Laskin et al., 2020; Nabati et al., 2023; Ren et al., 2023). Our work offers a unified framework based on low Inherent Bellman Error (IBE) and spectral theory (Ren et al., 2023). We define IBE below.

**Definition 2** (Inherent Bellman Error (IBE), (Zanette et al., 2020a))**.** *Given an MDP and a feature map $\phi$, the Inherent Bellman Error (IBE) is defined as:*

$$\mathcal{I}_\phi := \sup_{Q \in \mathcal{Q}_\phi} \inf_{\tilde{Q} \in \mathcal{Q}_\phi} \left\|\mathcal{T}Q - \tilde{Q}\right\|_\infty = \sup_{\theta \in \mathcal{B}_\phi} \inf_{\tilde{\theta} \in \mathcal{B}_\phi} \left\|\mathcal{T}Q_\theta - Q_{\tilde{\theta}}\right\|_\infty.$$

If $\mathcal{I}_\phi = 0$, $\mathcal{Q}_\phi$ is closed under $\mathcal{T}$ (up to projection onto $\mathcal{B}_\phi$). $\mathcal{I}_\phi$ quantifies how well $\mathcal{T}$ maps $\mathcal{Q}_\phi$ to itself. Zero IBE means any $\mathcal{T}Q_\theta$ for $Q_\theta \in \mathcal{Q}_\phi$ is perfectly representable by some $Q_{\tilde{\theta}} \in \mathcal{Q}_\phi$. This generalizes Linear MDPs (Jin et al., 2020), where $\mathcal{T}$ maps any Q-function into $\mathcal{Q}_\phi$.

## 2.2 Facilitating Efficient Exploration with Low-IBE Features

A representation with zero or low IBE facilitates efficient exploration. With expected feature $\phi_\pi = \mathbb{E}_{(s,a)\sim d^\pi}[\phi(s,a)]$ under policy $\pi$ (occupancy $d^\pi$), effective exploration aims to reduce uncertainty, e.g., by minimizing $\|\phi_\pi\|_{\Sigma^{-1}}$ across policies. Zanette et al. (2020b) define maximum uncertainty:

**Definition 3** (Max Uncertainty (Zanette et al., 2020b)). $\mathcal{U}(\sigma) := \max_{\pi, \|\xi\|_\Sigma \leq \sqrt{\sigma}} \phi_\pi^\top \xi := \max_\pi \sqrt{\sigma} \|\phi_\pi\|_{\Sigma^{-1}}$

Informally, our goal is to reduce maximum uncertainty. This can be done by overestimating it via sampling exploration noise $\xi \in \mathbb{R}^d \sim \mathcal{N}(0, \sigma\Sigma^{-1})$ per rollout (i.e., Thompson Sampling), where $\|\xi\|_2 \leq \frac{R_{max}}{1-\gamma}$ (Zanette et al., 2020b).

Motivated by the IBE, in what follows we detail a method for learning low-IBE representations and integrate them with exploration methods like Thompson Sampling (TS) to improve RL performance.

## 3 Spectral Bellman Method

This section introduces the Spectral Bellman Method (SBM), a framework designed to learn representations that satisfy the zero-IBE condition (Definition 2). While the zero-IBE condition implies that the function space is closed under the Bellman operator, directly enforcing this condition leads to computationally intractable min-max optimization problems. Our analysis reveals a fundamental connection between the spectral properties of the optimal Bellman operator and the covariance structure of the learned features under orthogonality assumptions. By leveraging this connection, inspired by the power iteration method, we derive a new learning objective that overcomes the limitations of direct Bellman error minimization.

### 3.1 Zero-IBE Objective

Consider the mapping $\tilde{\theta}(\theta) : \mathcal{B}_\phi \to \mathcal{B}_\phi$, that maps a parameter $\theta \in \mathcal{B}_\phi$ into his correspond Bellman minimizer $\tilde{\theta} \in \mathcal{B}_\phi$ (noting that for the optimal Q-function parameter, this is the identity map):

$$\tilde{\theta}(\theta) := \underset{\tilde{\theta} \in \mathcal{B}_\phi}{\arg\min} \|\mathcal{T}Q_\theta - Q_{\tilde{\theta}}\|_\infty$$

Our goal is to learn features $\phi(s,a) \in \mathbb{R}^d$ and a corresponding mapping $\tilde{\theta}(\theta)$ such that the function space $\mathcal{Q}_\phi$ is closed under the Bellman operator, i.e., $\mathcal{I}_\phi = 0$. However, a straightforward idea for minimizing the IBE leads to a complicated optimization (Modi et al., 2024), and thus, difficult to be implemented for practical applications.

Given a set of observations $\{s_i, a_i, \theta_i\}_{i=1}^N$ sampled from distributions $\rho(s,a)$ over state-action pairs and $\nu(\theta)$ over input Q-function parameters, IBE condition implies the minimization of mean squared error (MSE) between the Bellman backup of $Q_\theta$ and its representation in the learned feature space:

$$\mathcal{L}_{MSE}(\phi, \tilde{\theta}) = \mathbb{E}_{(s,a)\sim\rho(s,a), \theta\sim\nu(\theta)}\left[\left\|\mathcal{T}Q_\theta(s,a) - \phi(s,a)^\top\tilde{\theta}(\theta)\right\|_2^2\right]. \tag{1}$$

However, even the minimization of Equation (1) is still challenging. First, the Bellman operator $\mathcal{T}$ is highly non-linear in $Q_\theta$ (which depends on the learned $\phi$ and $\theta$) and the task parameters $\theta$, complicating the joint optimization landscape and risking suboptimal minima or poor features. Second, this MSE objective neither leverages the underlying spectral properties of $\mathcal{T}Q_\theta$ nor inherently promotes desirable structural properties in $\phi$ and $\tilde{\theta}$. To address these limitations, we next introduce a spectral analysis of the Bellman operator to derive a more principled learning objective.

The core distinction of SBM is its objective derived from the zero IBE condition, which seeks closure of $\mathcal{Q}_\phi$ under the Bellman operator across a *distribution* of parameters. This contrasts with: (1) methods focusing only on the optimal Q-function (Mnih et al., 2013; Kapturowski et al., 2018), which may be unfeasible to learn directly (Du et al., 2019); (2) Learn a successor representation or Laplace

---

**Algorithm 1** Spectral Bellman Power Method

---

1: Initialize $\tilde{\theta}_0, \phi_0$
2: **for** $t = \{1, 2, \ldots\}$ **do**
3:    Compute $\Lambda_{1,t} = \mathbb{E}_{\rho(s,a)}\left[\phi_t(s,a)\phi_t(s,a)^\top\right], \quad \Lambda_{2,t} = \mathbb{E}_{\nu(\theta)}\left[\tilde{\theta}_t\tilde{\theta}_t^\top\right].$
4:    Find $\phi$ and $\tilde{\theta}$ which satisfy the following constraints:
$$\Lambda_{2,t}\bar{\phi}(s,a) = \langle \overline{\mathcal{TQ}}_\cdot(s,a), \bar{\theta}_t(\cdot)\rangle, \quad (s,a) \in \mathcal{S} \times \mathcal{A}$$
$$\Lambda_{1,t}\bar{\theta}(\theta) = \langle \overline{\mathcal{TQ}}_\theta(\cdot), \bar{\phi}_t(\cdot)\rangle, \quad \theta \in \mathcal{B}_{\phi_t}$$
$$\langle \bar{\phi}_i(\cdot), \bar{\phi}_j(\cdot)\rangle = 0, \quad \langle \bar{\theta}_i(\cdot), \bar{\theta}_j(\cdot)\rangle = 0, \quad i \neq j \in [d]. \tag{3}$$
5: **end for**

---

methods (Mahadevan, 2005; Barreto et al., 2017; Touati & Ollivier, 2021; Farebrother et al., 2023), which are task-agnostic and not task-specific as our method.

## 3.2 BELLMAN SPECTRAL DECOMPOSITION UNDER ZERO IBE

To overcome the challenges of directly minimizing Equation (1), we analyze the spectral properties of the optimal Bellman operator under the zero-IBE condition. This analysis will lead us to an algorithm inspired by the power iteration method for singular value decomposition (Golub & Van Loan, 2013; Trefethen & Bau, 2022), designed to efficiently learn a zero-IBE representation.

For conceptual clarity, let us temporarily adopt a matrix perspective, assuming finite state-action spaces ($|\mathcal{S} \times \mathcal{A}| = n$) and a finite parameter space ($|\mathcal{B}_\phi| = m$). Let $\kappa : [n] \to \mathcal{S} \times \mathcal{A}$ be a bijection from indices to state-action pairs. We denote the feature matrix as $\Phi \in \mathbb{R}^{n \times d}$ (where the $i$-th row is $\phi(\kappa(i))^\top$) and the matrix of input parameters as $\Theta = [\theta_1, \ldots, \theta_m] \in \mathbb{R}^{d \times m}$. A Q-function $Q_\theta$ is then $\Phi\theta$, and the collection of Q-functions for all $\theta_j \in \Theta$ is $Q = \Phi\Theta \in \mathbb{R}^{n \times m}$. Under the zero-IBE condition, for every $\theta \in \mathcal{B}_\phi$, there exists a $\tilde{\theta}(\theta)$ such that $\mathcal{T}(\Phi\theta) = \Phi\tilde{\theta}(\theta)$. In matrix form, this means $\mathcal{TQ} = \Phi\tilde{\Theta}$, where $\tilde{\Theta} = [\tilde{\theta}(\theta_1), \ldots, \tilde{\theta}(\theta_m)] \in \mathbb{R}^{d \times m}$. To incorporate the influence of the sampling distributions $\rho(s,a)$ and $\nu(\theta)$, we augment the feature and parameter mappings:
$$\bar{\phi}(s,a) := \sqrt{\rho(s,a)}\phi(s,a), \quad \bar{\theta}(\theta) := \tilde{\theta}(\theta)\sqrt{\nu(\theta)}. \tag{2}$$
Consequently, the Bellman operator acting on a distribution of Q-functions can be thought of through an augmented operator, $\overline{\mathcal{TQ}}_\theta(s,a) = \bar{\phi}(s,a)^\top \bar{\theta}(\theta)$. In essence, the augmentation defined in Equation (2) enables us to treat the distribution-weighted problem as a standard spectral decomposition on the augmented space.

We also define the feature covariance matrix $\Lambda_1 := \mathbb{E}_{(s,a)\sim\rho(s,a)}\left[\phi(s,a)\phi(s,a)^\top\right]$ and the post-Bellman parameter covariance matrix $\Lambda_2 := \mathbb{E}_{\theta\sim\nu(\theta)}\left[\tilde{\theta}(\theta)\tilde{\theta}(\theta)^\top\right]$.

A key insight reveals a deep structural connection under the zero-IBE condition. Informally, when the Bellman operator $\mathcal{T}$ transforms a distribution of Q-functions (weighted by $\rho$ and $\nu$), the resulting (weighted) feature matrix $\Phi$ and (weighted) post-Bellman parameter matrix $\tilde{\Theta}$ correspond to the singular vectors of this transformation.

Our spectral Bellman method aim to find a representation that is aligned with the singular vectors of the (augmented) Bellman operator. Namely, $\Lambda_1$ and $\Lambda_2$ are diagonal matrices, correspond to the singular values of the Bellman operator. This structural alignment is crucial and forms the basis of our method. We refer the reader to Section A for a formal statement and proof of this connection.

This spectral relationship naturally motivates an approach analogous to the power iteration method for finding dominant eigenvectors/singular vectors (Golub & Van Loan, 2013; Trefethen & Bau, 2022). The following proposition captures identities central to such an iterative process (see Section B for proof).

**Proposition 1.** *The following identities hold:*
$$\langle \overline{\mathcal{TQ}}_\theta(\cdot), \bar{\phi}(\cdot)\rangle = \Lambda_1\bar{\theta}(\theta), \qquad \langle \overline{\mathcal{TQ}}_\cdot(s,a), \bar{\theta}(\cdot)\rangle = \Lambda_2\bar{\phi}(s,a).$$

The above identities closely resemble the update rules of a power iteration algorithm (Sanger, 1988; Xie et al., 2015; Guo et al., 2025), but for eigenfunction learning for symmetric operator in semi-supervised learning. This suggests an alternating optimization strategy for learning $\phi$ and $\tilde{\theta}$. Algorithm 1 outlines this iterative procedure. In each round, covariance matrices are computed,

followed by solving linear equations to extract the updated representation $\phi$ and parameters $\tilde{\theta}$. This spectral method offers significant advantages over direct minimization MSE loss (Equation (1)), as we will discuss in Section 3.4. The alternating updates can stabilize the optimization by decomposing the problem into simpler, often convex, subproblems. Moreover, by leveraging the power iteration structure, our method aims for the faster convergence rates characteristic of such techniques, leading more efficiently to the desired representation.

### 3.3 PRACTICAL SBM LEARNING

While Algorithm 1 provides a conceptual framework based on power iteration, directly solving the system of linear equations in Equation (3) at each step can be computationally prohibitive, especially in environments with large or infinite state-action spaces. To make this approach practical, we formulate an objective function whose minimization effectively performs these power iteration steps.

Let $\Lambda_{1,t} = \mathbb{E}_{\rho(s,a)}\left[\phi_t(s,a)\phi_t(s,a)^\top\right]$ and $\Lambda_{2,t} = \mathbb{E}_{\nu(\theta)}\left[\tilde{\theta}_t(\theta)\tilde{\theta}_t(\theta)^\top\right]$ denote the empirical covariance matrices at iteration $t$. The SBM objective is given by

$$\mathcal{L}(\phi, \tilde{\theta}; \nu, \rho) = \mathcal{L}_1(\phi) + \mathcal{L}_2(\tilde{\theta}) \quad s.t \quad \phi \in \mathcal{M}^\rho_{\mathcal{S}\times\mathcal{A}}, \tilde{\theta} \in \mathcal{M}^\nu_{\mathcal{B}_\phi} \qquad \text{(SBM Loss)}$$

where $\mathcal{L}_1(\phi) = \mathbb{E}_{\nu(\theta)\rho(s,a)}\left[\|\phi(s,a)\|^2_{\Lambda_{2,t}} - 2\mathcal{T}Q_{\theta,t}(s,a)\phi(s,a)^\top\tilde{\theta}_t(\theta)\right]$ is the representation loss; $\mathcal{L}_2(\tilde{\theta}) = \mathbb{E}_{\nu(\theta)\rho(s,a)}\left[\|\tilde{\theta}(\theta)\|^2_{\Lambda_{1,t}} - 2\mathcal{T}Q_{\theta,t}(s,a)\tilde{\theta}(\theta)^\top\phi_t(s,a)\right]$ is the parameter objective; and

$$\mathcal{M}^\mathbb{P}_\mathcal{X} = \{f : \mathcal{X} \to \mathbb{R}^d \quad | \quad \mathbb{E}_{x\sim\mathbb{P}}[f_i(x)f_j(x)] = 0 \quad \forall i \neq j \in [d]\}$$

is the orthogonal function space of functions over $\mathcal{X}$ w.r.t probability $\mathbb{P} : \mathcal{X} \to [0,1]$.

The term $\mathcal{L}_1$ updates $\phi$ to align with the Bellman-transformed Q-values, using the current estimate of parameter covariance $\Lambda_{2,t}$ and parameters $\tilde{\theta}_t$. The term $\mathcal{L}_2$ updates $\tilde{\theta}$ to best represent the Bellman-transformed Q-values given the current features $\phi_t$ and their covariance $\Lambda_{1,t}$. Indeed, minimizing the objective in SBM Loss is equivalent to applying the power method. This is formally shown by the following proposition.

**Proposition 2.** *For any $t$, a solution $\theta_t^*, \phi_t^*$ to the power method objective in Equation (3) if and only if it is a minimizer of the SBM Loss.*

A proof is provided in Section C. This equivalence means that by minimizing $\mathcal{L}(\phi, \tilde{\theta})$ using gradient-based methods, we are effectively performing the updates of the power iteration algorithm. In practice, we solve an unconstrained relaxed version of SBM Loss (see Section E for details).

### 3.4 COMPARISON OF THE SBM LOSS TO THE BELLMAN MSE

To better understand the advantages of the SBM Loss over the naive MSE objective (Equation (1)), consider the following expansion of the MSE objective:

$$\min_{\phi,\tilde{\theta}} \mathbb{E}_{(s,a)\sim\rho(s,a),\theta\sim\nu(\theta)}\left[\left\|\mathcal{T}Q_\theta(s,a) - \phi(s,a)^\top\tilde{\theta}(\theta)\right\|^2_2\right]$$
$$= \min_{\phi,\tilde{\theta}} \mathbb{E}_{(s,a)\sim\rho(s,a),\theta\sim\nu(\theta)}\left[C - 2\mathcal{T}Q_\theta(s,a)\phi(s,a)^\top\tilde{\theta}(\theta) + \phi(s,a)^\top\tilde{\theta}(\theta)\tilde{\theta}(\theta)^\top\phi(s,a)\right]$$
$$\propto \min_{\phi,\tilde{\theta}} \mathbb{E}_{(s,a)\sim\rho(s,a),\theta\sim\nu(\theta)}\left[\|\phi(s,a)\|^2_{\hat{\Lambda}} - 2\mathcal{T}Q_\theta(s,a)\phi(s,a)^\top\tilde{\theta}(\theta)\right],$$

where $C$ is a constant independent of $\phi$ and $\tilde{\theta}$, and $\hat{\Lambda} = \tilde{\theta}(\theta)\tilde{\theta}(\theta)^\top$ is a noisy, single-sample estimate of the parameter covariance. The SBM objective offers distinct advantages. First, its separation into $\mathcal{L}_1(\phi)$ and $\mathcal{L}_2(\tilde{\theta})$ enables alternating optimization (inherent to the power method), decomposing the joint problem into tractable subproblems and promoting stability over simultaneous MSE updates. Second, SBM's quadratic terms ($\|\phi(s,a)\|^2_{\Lambda_{2,t}}$, $\|\tilde{\theta}(\theta)\|^2_{\Lambda_{1,t}}$) use moving average covariance matrices $\Lambda_{2,t}, \Lambda_{1,t}$ from the previous iteration, offering robust, statistically grounded regularization. In contrast, MSE's quadratic term $\|\phi(s,a)\|^2_{\hat{\Lambda}}$ uses the noisy, single-sample $\hat{\Lambda}$, lacking this stabilizing batch-averaged influence. Finally, SBM incorporates an explicit orthogonality regularizer $\mathcal{L}_{orth}$.

These differences demonstrate SBM's more refined approach to representation learning over direct MSE minimization. These fundamental differences highlight how SBM offers a more refined approach

---

**Algorithm 2** Q-Learning with SBM

---

1: Initialize Q-function parameters $\hat{\theta}_0$, features $\phi_0$, and replay buffer $\mathcal{D}$.
2: **for** $t = 1, 2, \ldots,$ **do**
3:     **Data Collection with TS Exploration:**
4:         Generate $\hat{\theta}_{TS} \sim \mathcal{N}(\hat{\theta}_t, \sigma_{exp}\Sigma_t^{-1})$ using current $\hat{\theta}_t$ and $\Sigma_t$ from $\phi_t$ (Equation (5)).
5:         Collect data using policy $\pi_{\hat{\theta}_{TS}}(s) = \arg\max_{a \in \mathcal{A}} \phi_t(s, a)^\top \hat{\theta}_{TS}$, and add to $\mathcal{D}$.
6:     **Policy Optimization:**
7:         Update Q-function parameters: $\hat{\theta}_{t+1} = \arg\min_\theta \mathcal{L}_{QL}(\theta; \phi_t)$ (Equation (6)), using $\mathcal{D}$.
8:     **Representation Learning:**
9:         Define $\nu_t(\theta) = \mathcal{N}(\hat{\theta}_{t+1}, \sigma_{rep}^2 I)$ and $\rho_t(s, a)$ as a probability over $\mathcal{D}$ (e.g. uniform).
10:         Update features: $\phi_{t+1} = \arg\min_\phi \mathcal{L}(\phi, \tilde{\theta}; \nu_t(\theta), \rho_t(s, a))$ (SBM Loss).
11: **end for**

---

for representation learning than direct MSE minimization, which is also justified empirically in (Guo et al., 2025).

## 3.5 Efficient Exploration using Thompson Sampling

After covering spectral representation learning part, a learned representation can be leveraged to establish an efficient exploration method. In this work, we focus on Thompson Sampling (TS) (Osband et al., 2013; 2016; Azizzadenesheli et al., 2018; Zanette et al., 2020b), which is particularly suited for settings with low-IBE representations. Indeed, given a dataset $\mathcal{D} = \{s_i, a_i, r_i, s_i'\}_{i=1}^N$, a least-square estimator for an optimal parameter $\theta^* \in \mathcal{B}_\phi$ (such that $\mathcal{T}Q_{\theta^*} = Q_{\theta^*}$) can be found:

$$\hat{\theta}_{LS} \in \arg\max_{\theta \in \mathcal{B}_\phi} \mathbb{E}_{(s,a) \sim \mathcal{D}} \big[ (\mathcal{T}Q_\theta(s, a) - \phi(s, a)^\top \theta)^2 \big], \tag{4}$$

We conduct exploration following (Zanette et al., 2020b). Specifically, the weights $\hat{\theta}_{TS}$ for the behavior policy $\pi_{\hat{\theta}_{TS}}(s) = \arg\max_{a \in \mathcal{A}} \phi(s, a)^\top \hat{\theta}_{TS}$, are sampled from a posterior distribution, often referred to as the uncertainty set. Given the representation, the weights are sampled from the posterior before each rollout according to:

$$\hat{\theta}_{TS} \sim \mathcal{N}(\hat{\theta}_{LS}, \sigma_{exp}\Sigma^{-1}); \quad \Sigma = \lambda I + \sum_{(s,a) \in \mathcal{D}} \phi(s, a)\phi(s, a)^\top, \tag{5}$$

for some positive $\lambda$. TS provides an easy-to-implement method that add an exploration noise to the least-squre estimate $\hat{\theta}_{LS}$ (i.e., $\hat{\theta}_{TS} = \hat{\theta}_{LS} + \xi$), which yields the sampling procedure in Equation (5) and reduce the maximum uncertainty in Definition 3. We emphasize although we mainly consider the TS for exploration, the learned spectral Bellman representation is also compatible UCB with the same covariance $\Sigma$ in equation 5, as discussed in (Zanette et al., 2020b; Modi et al., 2024; Zhang et al., 2022). Having established the representation learning and exploration framework, we next explore its practical integration into value-based reinforcement learning algorithms, using Q-learning (Sutton et al., 1998) as an illustrative example due to its intrinsic connection with the optimal Bellman operator.

## 4 Q-Learning with SBM

Put the representation learning and RL together, we obtain Algorithm 2, which describes the implementation of the Spectral Bellman Method (SBM) using Q-learning and Thompson Sampling (TS) for exploration. The algorithm iteratively refines the policy and representation through three key phases: **i)** data collection with TS exploration, **ii)** policy optimization, and **iii)** spectral representation learning.

Algorithm 2 begins by initializing the Q-function parameters $\hat{\theta}_0$, feature mapping $\phi_0$, and an empty replay buffer $\mathcal{D}$ (Line 1). For every iteration $t$, the algorithm samples exploratory Q-function parameters $\hat{\theta}_{TS}$ (Lines 3-5) using the current Q-parameters $\hat{\theta}_t$ (as the mean for the TS posterior, akin to $\hat{\theta}_{LS}$ in Equation (4)) and the feature covariance $\Sigma_t$ (derived from the current features $\phi_t$ as per Equation (5)). The agent then interacts with the environment using a policy $\pi_{\hat{\theta}_{TS}}$ that is greedy with respect to $Q_{\hat{\theta}_{TS}}(s, a) = \phi_t(s, a)^\top \hat{\theta}_{TS}$. The collected state-action-reward transitions are stored in the

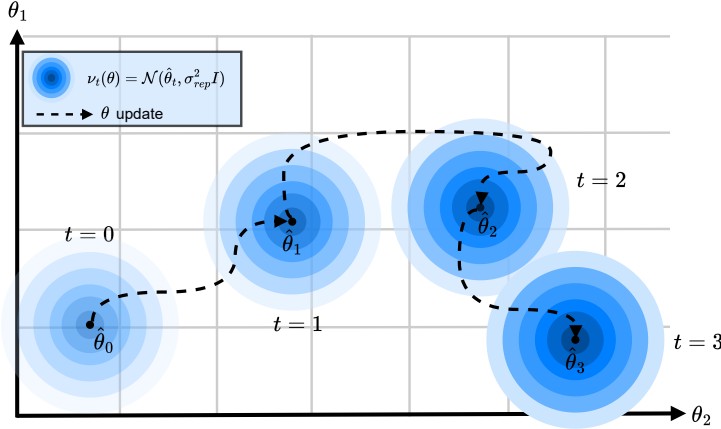

Figure 2: Visualization of the parameter sampling distribution $\nu_t(\theta) = \mathcal{N}(\hat{\theta}_t, \sigma^2_{rep}I)$ for $d = 2$ over successive rounds. As Q-learning updates the mean $\hat{\theta}_t$, $\nu(\theta)$ shifts, focusing representation learning on parameters relevant to the current policy. Darker regions indicate higher probability density.

replay buffer $\mathcal{D}$. This mechanism ensures that data collection is guided by the uncertainty captured in the learned feature space. Following data collection, the Q-function parameters are updated from $\hat{\theta}_t$ to $\hat{\theta}_{t+1}$ (Lines 6-7). This update is achieved by minimizing the standard loss with Q-learning target:

$$\mathcal{L}_{QL}(\theta; \phi) = \mathbb{E}_{(s,a)\sim\mathcal{D}}\left[(\mathcal{T}Q_{\theta^-}(s,a) - \phi(s,a)^\top\theta)^2\right], \tag{6}$$

where $\theta^-$ are the target parameters. This loss utilize the current features $\phi_t$ and mini-batches of data sampled from the replay buffer $\mathcal{D}$. This step refines the agent's policy based on the existing representation of state-action values.

Finally, in the final representation learning phase (Lines 8-10), the feature mapping is updated from $\phi_t$ to $\phi_{t+1}$ by minimizing the SBM Loss. A critical aspect of this phase is the choice of the distribution over parameters, $\nu(\theta)$ and over state-actions $\rho(s, a)$ Equation (2). $\nu(\theta)$ is centered around the newly updated Q-parameters obtained from the policy optimization phase: $\nu(\theta) = \mathcal{N}(\hat{\theta}_{t+1}, \sigma^2_{rep}I)$, where $\sigma^2_{rep}$ is a variance hyperparameter. The state-action distribution $\rho(s, a)$ for the SBM loss is implicitly defined by sampling transitions from the replay buffer $\mathcal{D}$. This adaptive focus of $\nu(\theta)$, as visualized in Figure 2, ensures that the feature learning process concentrates on satisfying the low-IBE condition in regions of the parameter space that are most relevant to the current policy.

This alternating process of exploration, policy optimization, and representation learning allows the policy and the features to co-evolve. Improved features lead to more accurate value estimates and consequently better policies. In turn, an improved policy guides the representation learning towards more effective exploration.

**Extension to Multi-Step Operators.** The SBM framework naturally extends to $h$-step Bellman operators $\mathcal{T}^h$. A low one-step IBE implies a low $h$-step IBE (Section D). Thus, we can apply the SBM Loss by simply replacing the one-step backup $\mathcal{T}Q_{\theta,t}$ with a multi-step target, such as one from Retrace($\lambda$) (Munos et al., 2016).

## 5 EXPERIMENTS

We evaluate the effectiveness of incorporating Spectral Bellman Representation learning into standard deep RL algorithms. We focus on challenging benchmarks and compare against established baselines.

### 5.1 SETTING

**Environments:** We use the Atari game suite (ALE, Bellemare et al. (2013)). We follow standard preprocessing steps, including frame stacking (4 frames), grayscale conversion, downsampling, and

| Method | Atari ALE | | Atari Explore | |
|---|---|---|---|---|
| | Mean | Median | Mean | Median |
| DQN (Mnih et al., 2013) | $1.62 \pm 0.12$ | 0.52 | $0.24 \pm 0.03$ | 0.11 |
| Online PVN ($\epsilon$-greedy) | $1.72 \pm 0.11$ | 0.60 | $0.26 \pm 0.03$ | 0.21 |
| Online PVN (TS) | $1.84 \pm 0.15$ | 0.65 | $0.31 \pm 0.04$ | 0.23 |
| SBM + DQN ($\epsilon$-greedy). | $1.80 \pm 0.13$ | 0.64 | $0.33 \pm 0.03$ | 0.23 |
| **SBM + DQN (TS)** | **$2.23 \pm 0.19$** | **0.85** | **$0.45 \pm 0.05$** | **0.24** |
| R2D2 (Kapturowski et al., 2018) | $3.21 \pm 0.22$ | 1.14 | $0.40 \pm 0.06$ | 0.22 |
| SBM + R2D2 ($\epsilon$-greedy) | $3.3 \pm 0.24$ | 1.14 | $0.45 \pm 0.05$ | 0.22 |
| **SBM + R2D2 (TS)** | **$3.53 \pm 0.23$** | **1.37** | **$0.61 \pm 0.03$** | **0.30** |

Table 1: Aggregated Atari HNS at 100M steps. Our SBM method with TS is in bold.

| Method | Atari ALE | Atari Explore |
|---|---|---|
| Features from DQN Loss (Azizzadenesheli et al., 2018) | $1.73 \pm 0.14$ | $0.30 \pm 0.03$ |
| Features from Naive MSE Loss | $1.82 \pm 0.12$ | $0.37 \pm 0.04$ |
| SBM w/o Orthogonality Reg. | $2.11 \pm 0.11$ | $0.43 \pm 0.05$ |
| SBM Full | $2.23 \pm 0.19$ | $0.45 \pm 0.05$ |

Table 2: Ablation study on the DQN backbone with TS exploration.

sticky actions unless specified otherwise. Performance is measured using mean and median Human Normalized Score (HNS) across all 55 games after 100 million environment steps.

To assess the impact of our method on exploration, we report aggregated HNS on a subset of particularly challenging Atari games identified by Badia et al. (2020). This subset includes games known for sparse rewards or requiring long-term credit assignment: *Montezuma's Revenge*, *Pitfall!*, *Private Eye*, *Skiing*, *Freeway*, *Solaris*, *Venture*, *BeamRider* and *Pong*. We denote this benchmark as Atari Explore.

**Implementation Details:** Our spectral learning method is integrated into two representative deep RL agents: DQN (Mnih et al., 2013) and R2D2 (Kapturowski et al., 2018).

We minimize the Equation (SBM Loss) using stochastic gradient descent. This is also true for the Q-learning phase. Therefore, SBM was actually tested in a fully online setting similar to DQN. To satisfy the theoretical assumption $\|\phi(s, a)\|_2 \leq 1$ used in the background (Section 2.1) , we apply norm clipping to the output of the feature encoder network. We follow Algorithm 2, performing alternating updates between the Q-learning objective (Equation (6)) and the spectral representation objective (SBM Loss). The distribution $\nu(\theta)$ is set to $\mathcal{N}(\hat{\theta}_{t+1}, \sigma_{rep}^2 I)$, as explained in Section 4. For specific hyperparameters and implementation details we refer the reader to Section E. We compare standard $\epsilon$-greedy exploration with the TS approach described in Section 3.5. As R2D2 operates in a distributed setting with multiple asynchronous actors, the precision matrix $\Sigma$ is shared across the actors while each of them sample $\hat{\theta}_{TS}$ before each epoch according to Equation (5). For R2D2 experiments, the spectral objective is applied using a multi-step operator in the form of Retrace operator target (Munos et al., 2016) in place of $\mathcal{T}Q_{\theta,t}$ (see Section D). We found that choosing $\sigma_{rep}$ is crucial for successful learning. We conducted a grid search and empirically found the best value to be $\sigma_{rep}^2 = 10^{-2}$.

In addition, in order to compare SBM to a Laplace based representation method, we compare SBM to PVN Farebrother et al. (2023). To fairly compare with SBM, we adapted PVN to our online learning setup. We alternate between policy optimization (DQN) and representation learning. The representation learning phase uses Random Network Indicators (RNI) to define auxiliary tasks, over the collected data, as in the original PVN. We used the same network architecture and latent dimension as our SBM experiments. We evaluated Online PVN with both $\epsilon$-greedy and TS. Lastly, we made an ablation study with a DQN backbone. We examine SBM with features extracted with naive MSE loss in Equation (1) and not with our suggested spectral representation. Furthermore, we investigated the specific case where $\nu_t(\theta) = \delta(\theta - \hat{\theta}_t)$ under the naive MSE loss. This configuration effectively corresponds to standard DQN training in Equation (6) augmented with TS exploration, suggest by Azizzadenesheli et al. (2018). We also examine the effect of the orthogonality regularization by training SBM without it. All runs are done over 10 seeds.

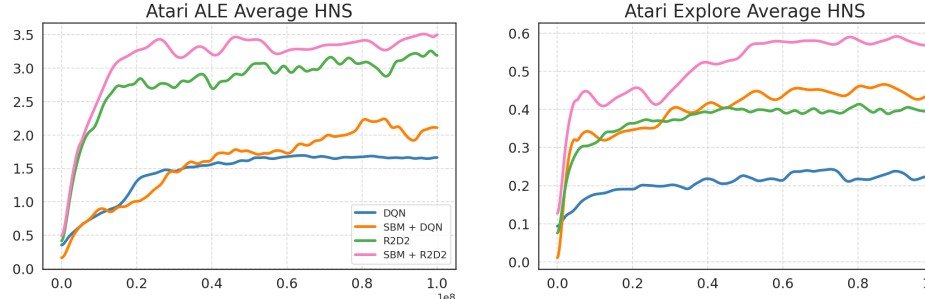

Figure 3: Average HNS over 100M steps. DQN and R2D2 against their SBM counterparts with TS across Atari ALE benchmark (left) and on the hard-exploration subset (right).

## 5.2 MAIN RESULTS

Table 1 presents results comparing the baseline agents with their SBM counterparts, which incorporate spectral representation learning and exploration. Figure 3 presents the results against environment steps over the training. Per game results are presented in Section F.

**DQN Comparison:** We find SBM significantly outperforms vanilla DQN. Furthermore, combining spectral learning with TS exploration yields substantial gains, particularly on hard exploration tasks, suggesting that the learned representation effectively facilitates structured exploration. We observed that SBM underperforms compared to DQN in precision-control games such as Breakout. This is a characteristic side-effect of optimistic exploration strategies (like TS) in environments with 'narrow' optimal paths or instant-death conditions, where high-variance actions often lead to termination. This effect can be mitigated by annealing the exploration noise or switching to greedy evaluation once a sufficient performance threshold is reached.

**R2D2 Comparison:** We augment the R2D2 agent, which utilizes recurrent networks and Retrace targets, with our spectral representation learning objective applied to the Retrace target. R2D2 with SBM demonstrates improved performance over the baseline R2D2, with the most significant improvement observed when combined with TS, particularly on the Atari Explore subset.

**Online PVN Comparison:** While Online PVN improves over DQN, SBM with TS performs better, especially on Atari Explore. This is likely because Online PVN learns reward-agnostic features using successor measure representation, capturing state-action structure on various MDPs, whereas SBM learns reward-dependent features by enforcing closure under the optimal Bellman operator. SBM's approach appears better for maximizing online rewards, while PVN's may be more suited for adapting to varied reward tasks.

**Ablation Study:** The results are presented in Table 2. SBM-learned features yielded significantly better results compared to features learned by minimizing the naive MSE loss, highlighting the advantage of the spectral objective over direct Bellman error minimization. While limiting representation learning to the current policy (effectively the DQN objective) combined with TS exploration improves performance over vanilla $\epsilon$-greedy DQN, it causes significant degradation compared to SBM or even the full Naive MSE loss. This indicates the critical importance of learning representations over a distribution of value functions rather than solely fitting the current policy. Additionally, performance without the orthogonality regularizer was slightly lower, suggesting that while beneficial, strict orthogonality is not the primary driver of SBM's gains. The computational overhead of SBM is relatively small, resulting in a $\sim 20\%$ increase in compute time per 1M steps. This overhead primarily stems from training the $\tilde{\theta}$ network and computing the inverse covariance for Thompson Sampling. Notably, the core representation learning cost mirrors vanilla baselines, while the Q-learning phase remains highly efficient by updating only linear weights.

## 6 RELATED WORK

Our work builds upon and distinguishes itself from several key areas in reinforcement learning.

**Representation Learning in RL:** Many auxiliary objectives, such as contrastive learning (Laskin et al., 2020; Schwarzer et al., 2021) and autoencoders (Kingma et al., 2019), don't explicitly optimize for Bellman compatibility. Laplace-based methods, including Successor Features (Mahadevan, 2005;

Barreto et al., 2017; Touati & Ollivier, 2021; Farebrother et al., 2023), often aim for task-agnostic, generalizable representations, potentially less potent for task-specific optimization as done in our work. Directly learning linear representations for the optimal Q-function can be sample inefficient under misspecification (Du et al., 2019; Azizzadenesheli et al., 2018). In contrast, SBM, derives its objective from structural properties under low IBE (Zanette et al., 2020a), targeting Bellman consistency across the function space, which makes the IBE-based model-free representation learning (Modi et al., 2024) eventually practical.

**Linear MDPs and IBE:** Linear MDPs (Jin et al., 2020; Yang & Wang, 2020) assume linear transitions and rewards in features, a practically challenging condition (Agarwal et al., 2020; Zhang et al., 2022; Ren et al., 2022). The IBE (Zanette et al., 2020a) relaxes this, measuring if the function space $\mathcal{Q}_\phi$ is nearly closed under the Bellman operator ($\mathcal{T}Q_\theta \approx Q_{\tilde{\theta}}$). Low IBE, a more direct target than full MDP linearization, ensures theoretical guarantees; e.g., Zanette et al. (2020a) presented a planning-based algorithm achieving $\tilde{\mathcal{O}}(dH^{1.5}\sqrt{T} + \sqrt{d}HT\mathcal{I}_\phi)$ regret under low-IBE settings. While practical methods approximate linear MDPs by combining representation learning for RL with some dynamics assumptions (Zhang et al., 2022; Ren et al., 2022; Shribak et al., 2024; Fujimoto et al., 2025), our work is inspired by the $\mathcal{I}_\phi = 0$ condition's structural implications, avoiding full Linear MDP decomposition which can be resource-intensive and require impractical latent dimensions with heuristic nonlinear correction in (Fujimoto et al., 2025).

**Bellman Error/Residual Minimization:** Traditional value-based RL minimizes the Bellman error via methods like gradient TD (Sutton et al., 2009), residual algorithms (Baird, 1995), and target networks (Mnih et al., 2013). Distributional RL (Bellemare et al., 2017) matches return distributions. Unlike these, our method doesn't directly minimize the residual for a single $\theta$, but leverages structural relationships (specifically covariance alignment) emerging from assuming zero residual across a function distribution induced by $\nu(\theta)$.

**Exploration Strategies:** In contrast to count-based methods (Bellemare et al., 2016) and intrinsic motivation (Pathak et al., 2017), TS offers a Bayesian uncertainty approach, effective in linear settings (Agrawal & Goyal, 2013; Azizzadenesheli et al., 2018). Notably, Zanette et al. (2020b) analyzes TS for RL with low-IBE representations. Our contribution is synergistic: our spectral objective learns suitable representations $\phi$, and TS naturally utilizes the resulting feature covariance $\Sigma_t$ for more directed exploration.

# 7 CONCLUSION

We introduced Spectral Bellman Method (SBM), a theoretically-motivated method for spectral representation learning and efficient TS based exploration. Our spectral objective derived from the zero-IBE condition allows for theoretically-grounded representation. We show that the SBM is compatible with multi-step operators such as Retrace. Experiments on Atari demonstrate consistent improvements over baselines, especially on hard-exploration games and when applied to advanced agents (R2D2), validating our approach.

A key limitation includes the sensitivity of SBM over the parameter distribution, which needs to be tuned carefully. Also, a broader empirical validation beyond Atari would be beneficial (e.g., continuous control tasks). Future work may focus on strengthening the theoretical analysis (convergence, non-zero IBE), developing improved algorithms (better approximations of $\tilde{\theta}$, adaptive parameter sampling), and extending the framework to other settings (e.g., distributional or actor-critic methods).

In conclusion, SBM offers a promising avenue for representation learning and exploration, potentially leading to more efficient, stable, and exploratory RL agents, by explicitly optimizing features for Bellman consistency across the function space.

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

## A  THE BELLMAN OPERATOR SPECTRAL DECOMPOSITION THEOREM

We start by incorporating distributions over states/actions and parameters, let $\rho(s, a)$ be a distribution over $\mathcal{S} \times \mathcal{A}$. In the finite case, let $P_{s,a}$ be a diagonal matrix with $\sqrt{\mathbb{P}(\kappa(i))}$ on the diagonal. Similarly, let $\nu_\theta$ represent weighting by $\sqrt{\nu(\theta)}$ (conceptually, an operator or a diagonal matrix if $\mathcal{B}_\phi$ and $m$ are finite). We define the augmented feature and parameter matrices as $\Phi_P = P_{s,a}\Phi$ and $\tilde{\Theta}_P = \tilde{\Theta}P_\theta$. The distribution-augmented Bellman matrix becomes:

$$\overline{\mathcal{T}Q} := P_{s,a}(\mathcal{T}Q)P_\theta = P_{s,a}\Phi\tilde{\Theta}P_\theta = \Phi_P\tilde{\Theta}_P.$$

The following theorem highlights a structural connection between the (weighted) feature matrix, (weighted) parameter matrix and the singular matrix of the Bellman matrix.

**Theorem 1** (Bellman Operator Spectral Decomposition). *Let $\phi : \mathcal{S} \times \mathcal{A} \to \mathbb{R}^d$ be a feature map with $\mathcal{I}_\phi = 0$. Let the feature covariance matrix under $\rho(s, a)$ be*

$$\Lambda := \mathbb{E}_{(s,a)\sim\rho(s,a)}\left[\phi(s,a)\phi(s,a)^\top\right] = \Phi_P^\top\Phi_P,$$

*where $\Lambda = diag(\{\lambda_i\}_{i=1}^d)$. Also, let the parameter covariance also be a diagonal matrix:*

$$M = \mathbb{E}_{\theta\sim\nu(\theta)}\left[\tilde{\theta}(\theta)\tilde{\theta}(\theta)^\top\right] = \tilde{\Theta}_P\tilde{\Theta}_P^\top,$$

*with $M = diag(\{\mu\}_{i=1}^d)$.*

*Consider the SVD of the augmented Bellman matrix $\overline{\mathcal{T}Q} = U\Sigma V^\top$, then the SVD components satisfy:*

1. *The non-zero singular values in $\Sigma$ (up to $d$) are related to the values of $\Lambda$ and $M$. Specifically, $\Sigma$ has the form $diag(\sqrt{\lambda_1\mu_1}, ..., \sqrt{\lambda_d\mu_d}, 0, ...)$ up to permutation. Let $\Sigma_d = diag(\sigma_1, \ldots, \sigma_d)$ where $\sigma_i = \sqrt{\lambda_i m_i}$.*

2. *The corresponding left singular vectors (first $d$ columns of $U$, denoted $\tilde{U}$) and right singular vectors (first $d$ columns of $V$, denoted $\tilde{V}$) satisfy:*

$$\Phi_P = \tilde{U}\Sigma_\phi \quad and \quad \tilde{\Theta}_P = \Sigma_\theta\tilde{V}^\top,$$

   *where $\Sigma_\phi\Sigma_\theta = \Sigma$, $\Sigma_\phi^2 = \Lambda_1$, and $\Sigma_\theta^2 = M$.*

*Proof.* Under the assumption of zero Bellman error, we have that

$$\overline{\mathcal{T}Q}^\top\overline{\mathcal{T}Q} = \tilde{\Theta}_P^\top\Phi_P^\top\Phi_P\tilde{\Theta}_P = V\Sigma^\top\Sigma V^\top$$

where the last equality follows from the SVD of $\overline{\mathcal{T}Q}$. Given that $\Phi_P^\top\Phi_P = \Lambda$ we have:

$$\tilde{\Theta}_P^\top\Lambda\tilde{\Theta}_P = V\Sigma^\top\Sigma V^\top.$$

Since we assume $M = \tilde{\Theta}_P\tilde{\Theta}_P^\top$ is diagonal, the rows of $\tilde{\Theta}_P$ are orthogonal. This implies that $\tilde{\Theta}_P$ can be factored as $\tilde{\Theta}_P = \Lambda_2^{1/2}\tilde{V}^\top$, where $\tilde{V} \in \mathbb{R}^{m\times d}$ has orthonormal columns. Substituting this into the expression:

$$(M^{1/2}\tilde{V}^\top)^\top\Lambda(M^{1/2}\tilde{V}^\top) = \tilde{V}M^{1/2}\Lambda M^{1/2}\tilde{V}^\top$$

Since $\Lambda$ and $M^{1/2}$ are diagonal, they commute. Thus:

$$\tilde{V}\Lambda M\tilde{V}^\top = V\Sigma^\top\Sigma V^\top$$

Comparing both sides, we identify that the singular values are $\Sigma^\top\Sigma = \Lambda M$ (implying the singular values are $\sqrt{\lambda_i m_i}$) and the right singular vectors correspond to $\tilde{V}$.

Similarly, for the left singular vectors, we have:

$$\overline{\mathcal{T}Q\mathcal{T}Q}^\top = \Phi_P\tilde{\Theta}_P\tilde{\Theta}_P^\top\Phi_P^\top = U\Sigma\Sigma^\top U^\top.$$

Using $\tilde{\Theta}_P\tilde{\Theta}_P^\top = M$ and the factorization $\Phi_P = \tilde{U}\Lambda^{1/2}$ (implied by $\Phi_P^\top\Phi_P = \Lambda$ and orthogonality of U):

$$(\tilde{U}\Lambda^{1/2})M(\tilde{U}\Lambda^{1/2})^\top = \tilde{U}(\Lambda M)\tilde{U}^\top.$$

This confirms that $U\Sigma_d = \tilde{U}(\Lambda M)^{1/2} = \tilde{U}\Lambda^{1/2}M^{1/2}$. Rearranging for the feature matrix $\Phi_P$ using the derived singular values yields the final relation:

$$\Phi_P = \tilde{U}\Sigma_d M^{-1/2} = \tilde{U}\Lambda^{1/2},$$

which completes the proof. □

## B   PROOF FOR PROPOSITION 1

*Proof.*

$$\langle \overline{\mathcal{T}Q}_\theta(\cdot), \bar{\phi}(\cdot)\rangle_{\rho(s,a)} = \int \bar{\phi}(s,a)\overline{\mathcal{T}Q}_\theta(s,a)dsda$$

$$= \int \phi(s,a)\phi(s,a)^\top \bar{\theta}(\theta)\rho(s,a)dsda$$

$$= \int \phi(s,a)\phi(s,a)^\top \rho(s,a)dsda\bar{\theta}(\theta)$$

$$= \Lambda_1 \tilde{\theta}(\theta).$$

$$\langle \overline{\mathcal{T}Q}_\cdot(s,a), \bar{\theta}(\cdot)\rangle_{\nu(\theta)} = \int \bar{\theta}(\theta)\overline{\mathcal{T}Q}_\theta(s,a)d\theta$$

$$= \int \tilde{\theta}(\theta)\bar{\phi}(s,a)^\top \tilde{\theta}(\theta)\nu(\theta)d\theta$$

$$= \int \tilde{\theta}(\theta)\tilde{\theta}(\theta)^\top \nu(\theta)d\theta\bar{\phi}(s,a)$$

$$= \Lambda_2 \bar{\phi}(s,a).$$

□

## C   PROOF FOR PROPOSITION 2

*Proof.* The equivalent objective for the $t$-th iteration of the power method is given by

$$\min_{\phi,\tilde{\theta}} \int \|\Lambda_{2,t}\phi(s,a) - \langle \mathcal{T}Q_\cdot(s,a), \tilde{\theta}_t(\cdot)\rangle\|^2 dsda + \int \|\Lambda_{1,t}\tilde{\theta}(\theta) - \langle \mathcal{T}Q_\theta(\cdot), \phi_t(\cdot)\rangle\|^2 d\theta \tag{7}$$

$$\text{s.t} \quad \mathbb{E}_{\rho(s,a)}[\phi_i(s,a)\phi_j(s,a)] = \mathbb{E}_{\nu(\theta)}\left[\tilde{\theta}_i(\theta)\tilde{\theta}_j(\theta)\right] = 0 \quad \forall i \neq j \in [d]$$

The choice of the norm is arbitrary. Therefore, we can choose the norm induced by $\Lambda_{1,t}^{-1}$ and $\Lambda_{2,t}^{-1}$ for the left and right terms respectively

$$\int \left\|\Lambda_{2,t}\bar{\phi}(s,a) - \int \overline{\mathcal{T}Q}_\theta(s,a)\bar{\theta}_t(\theta)d\theta\right\|^2_{\Lambda_{2,t}^{-1}} dsda$$

$$= \int \{C_1 + \bar{\phi}(s,a)^\top \Lambda_{2,t}\bar{\phi}(s,a) - 2\bar{\phi}(s,a)^\top \int \overline{\mathcal{T}Q}_\theta(s,a)\bar{\theta}_t(\theta)d\theta\}dsda$$

$$= \int C_1 \, dsda + \mathbb{E}_{\rho(s,a)}\left[\phi(s,a)^\top \Lambda_{2,t}\phi(s,a)\right] - 2\mathbb{E}_{\nu(\theta)\rho(s,a)}\left[\mathcal{T}Q_{\theta,t}(s,a)\phi(s,a)^\top \tilde{\theta}_t(\theta)\right]$$

where $C_1$ is a term that does not depend on $\phi$. The same practice can be used for the right term in Eq 7

$$\mathbb{E}_{\nu(\theta)}\left[\tilde{\theta}(\theta)^\top \Lambda_{1,t}\tilde{\theta}(\theta)\right] - 2\mathbb{E}_{\nu(\theta)\rho(s,a)}\left[\mathcal{T}Q_{\theta,t}(s,a)\tilde{\theta}(\theta)^\top \phi_t(s,a)\right] + C_2$$

where $C_2$ is a constant which does not depend on $\tilde{\theta}$. Therefore, the overall equivalent objective is

$$\mathcal{L}(\phi,\theta;\nu,\rho) = \mathcal{L}_1(\phi) + \mathcal{L}_2(\tilde{\theta}) \quad s.t \quad \phi \in \mathcal{M}^\rho_{\mathcal{S}\times\mathcal{A}}, \tilde{\theta} \in \mathcal{M}^\nu_{\mathcal{B}_\phi}$$

where

$$\mathcal{L}_1(\phi) = \mathbb{E}_{\rho(s,a)}\left[\phi(s,a)^\top \Lambda_{2,t}\phi(s,a)\right] - 2\mathbb{E}_{\nu(\theta)\rho(s,a)}\left[\mathcal{T}Q_{\theta,t}(s,a)\phi(s,a)^\top \tilde{\theta}_t(\theta)\right].$$

$$\mathcal{L}_2(\tilde{\theta}) = \mathbb{E}_{\nu(\theta)}\left[\tilde{\theta}(\theta)^\top \Lambda_{1,t}\tilde{\theta}(\theta)\right] - 2\mathbb{E}_{\nu(\theta)\rho(s,a)}\left[\mathcal{T}Q_{\theta,t}(s,a)\tilde{\theta}(\theta)^\top \phi_t(s,a)\right].$$

□

For the practical unconstrained version of SBM Loss see Section E.

## D    MULTI-STEP ALGORITHMS

### D.1    THE $h$-STEP OPTIMAL BELLMAN OPERATOR

While the analysis in Section 3 focused on the standard one-step Bellman operator $\mathcal{T}$, the spectral representation learning framework can be naturally extended to multi-step operators. Multi-step methods are widely used in modern RL agents as they often accelerate learning by propagating information more rapidly through trajectories (Sutton et al., 1998; Pohlen et al., 2018; Schrittwieser et al., 2020). Consider the $h$-step optimal Bellman operator, defined recursively:

$$\mathcal{T}^h Q(s,a) = \mathcal{T}(\mathcal{T}^{h-1}Q)(s,a) = r(s,a) + \gamma \mathbb{E}_{s' \sim P(\cdot|s,a)} \left[ \max_{a' \in \mathcal{A}} \mathcal{T}^{h-1}Q(s',a') \right]. \tag{8}$$

The optimal Q-function $Q^*$ is also the fixed point of $\mathcal{T}^h$ for any $h \geq 1$ (Efroni et al., 2018). Furthermore, $\mathcal{T}^h$ is known to be a contraction mapping with factor $\gamma^h$ in the max-norm (Efroni et al., 2018), which is potentially beneficial for stability. Evaluating $\mathcal{T}^h Q$ generally requires access to a model of the environment, either through a simulator or a learned model (Schrittwieser et al., 2020), often approximated using planning techniques like Monte Carlo Tree Search (MCTS).

We can define the inherent Bellman error with respect to this $h$-step operator:

$$\mathcal{I}_\phi^h := \sup_{\theta \in \mathcal{B}} \inf_{\tilde{\theta} \in \mathcal{B}} \left\| \mathcal{T}^h Q_\theta - Q_{\tilde{\theta}} \right\|_\infty. \tag{9}$$

Crucially, if the function space $\mathcal{Q}_\phi$ is closed under the one-step operator $\mathcal{T}$, it is also closed under $\mathcal{T}^h$.

**Proposition 3** ($h$-step IBE Bound). *Consider a feature map $\phi$. If the one-step IBE is $\mathcal{I}_\phi$, then the $h$-step IBE satisfies $\mathcal{I}_\phi^h \leq \sum_{i=0}^{h-1} \gamma^i \mathcal{I}_\phi \leq \frac{1}{1-\gamma} \mathcal{I}_\phi$. In particular, $\mathcal{I}_\phi = 0$ iff $\mathcal{I}_\phi^h = 0$.*

*Proof.* Using the definition $\mathcal{I}_\phi = \sup_Q \inf_{\tilde{Q}} \|\mathcal{T}Q - \tilde{Q}\|_\infty$, there exists $\tilde{Q}_1$ such that $\|\mathcal{T}Q - \tilde{Q}_1\|_\infty \leq \mathcal{I}_\phi$. Similarly, there exists $\tilde{Q}_2$ such that $\|\mathcal{T}\tilde{Q}_1 - \tilde{Q}_2\|_\infty \leq \mathcal{I}_\phi$. Then, using the triangle inequality and the $\gamma$-contraction property of $\mathcal{T}$:

$$\begin{aligned}
\|\mathcal{T}^2 Q - \tilde{Q}_2\|_\infty &= \|\mathcal{T}(\mathcal{T}Q) - \mathcal{T}\tilde{Q}_1 + \mathcal{T}\tilde{Q}_1 - \tilde{Q}_2\|_\infty \\
&\leq \|\mathcal{T}(\mathcal{T}Q) - \mathcal{T}\tilde{Q}_1\|_\infty + \|\mathcal{T}\tilde{Q}_1 - \tilde{Q}_2\|_\infty \\
&\leq \gamma\|\mathcal{T}Q - \tilde{Q}_1\|_\infty + \mathcal{I}_\phi \leq \gamma\mathcal{I}_\phi + \mathcal{I}_\phi.
\end{aligned}$$

Repeating this argument $h$ times yields the bound $\mathcal{I}_\phi^h \leq \sum_{i=0}^{h-1} \gamma^i \mathcal{I}_\phi$. $\qquad \square$

This proposition implies that if features $\phi$ yield low (or zero) one-step IBE, they also yield low (or zero) $h$-step IBE. Therefore, the spectral representation learning objective in SBM Loss can be directly applied by replacing the one-step operator $\mathcal{T}Q_{\theta,t}$ with its $h$-step counterpart $\mathcal{T}^h Q_{\theta,t}$ (or its approximation, e.g., via MCTS). This encourages learning features that linearly represent the outcome of $h$-step lookahead planning.

### D.2    PRACTICAL MULTI-STEP TARGETS

While $\mathcal{T}^h$ offers theoretical appeal, practical deep RL algorithms often employ sampled multi-step targets derived from actual trajectories, such as $n$-step Q-learning targets or Retrace (Munos et al., 2016). Let $Q_\theta(s,a) = \phi(s,a)^\top \theta$ and $\pi_\theta(s) = \arg\max_a Q_\theta(s,a)$ (such that $\mathcal{T}^{\pi_\theta} Q_\theta = \mathcal{T}Q_\theta$). A generic $n$-step target involves bootstrapping off $Q_\theta$ after $n$ steps.

However, using such targets within our spectral learning framework, which involves sampling different $\theta \sim \nu(\theta)$ poses a challenge in off-policy settings. The target policy $\pi_\theta$ associated with each sampled $\theta$ will likely differ significantly from the behavior policy $\mu$ used to generate the data in the replay buffer $\mathcal{D}$. This discrepancy can lead to high variance or instability, especially for longer multi-step returns.

To mitigate this, off-policy correction techniques like Retrace($\lambda$) (Munos et al., 2016) are crucial. Retrace provides a return target that mixes $n$-step returns and function approximation values, using

importance sampling ratios $c_k = \beta \min\{1, \frac{\pi_\theta(a_k|s_k)}{\mu(a_k|s_k))}\}$:

$$\mathcal{R}^\beta Q_\theta(s_t, a_t) = Q_\theta(s_t, a_t) + \mathbb{E}_\mu\left[\sum_{k=t}^\infty \gamma^{k-t}\left(\prod_{j=t+1}^k c_j\right)\delta_k\right], \tag{10}$$

where $\delta_k = \mathcal{T}^{\pi_\theta}Q_\theta(s_k, a_k) - Q_\theta(s_k, a_k)$ is the TD error. In practice, the used target policy $\pi_\theta^{\epsilon_t}$ is an $\epsilon$-greedy version of $\pi_\theta$ with $\epsilon_t$ and $t \to 0$ along the training process such that $\|\mathcal{T}^{\pi_\theta^{\epsilon_t}}Q_\theta - \mathcal{T}Q_\theta\|_\infty \leq \epsilon_t\|Q_\theta\|_\infty$ as suggested by (Munos et al., 2016).

Furthermore, to handle varying reward scales across different environments (e.g., in Atari), value transformations are often applied (Pohlen et al., 2018). A common transformation is $f(x) = \text{sign}(x)(\sqrt{|x|+1} - 1) + \epsilon x$ for small $\epsilon$. The target operator becomes $\mathcal{R}_f^\beta Q = f(\mathcal{R}^\beta f^{-1}(Q))$.

While the strict condition $\mathcal{I}_\phi = 0$ does not guarantee that $\mathcal{R}^\beta Q_\theta$ lies within $\mathcal{Q}_\phi$, we can still hypothesize that learning features $\phi$ to linearly represent these practical, information-rich targets is beneficial. We can heuristically apply SBM Loss by replacing $\mathcal{T}Q_{\theta,t}$ with $\mathcal{R}^\beta Q_{\theta,t}$. This encourages $\phi$ to capture structure relevant to these robust, multi-step, off-policy corrected targets used in high-performing agents like R2D2 (Kapturowski et al., 2018), even if the direct IBE connection is relaxed.

# E   IMPLEMENTATION DETAILS: SBM

## E.1   UNCONSTRAINED LOSS FOR SBM LOSS

The unconstrained objective (the Lagrangian) of SBM Loss is:
$$\mathcal{L}(\phi, \theta; \nu, \rho) = \mathcal{L}_1(\phi) + \mathcal{L}_2(\tilde{\theta}) + \mathcal{L}_{orth}(\phi, \tilde{\theta}),$$

$\mathcal{L}_{orth}(\phi, \tilde{\theta}) = \mathbb{E}_{\theta \sim \nu(\theta)}\left[\sum_{i \neq j \in [d]} \lambda_{i,j}\tilde{\theta}_i(\theta)\tilde{\theta}_j(\theta)\right] + \mathbb{E}_{(s,a) \sim \rho(s,a)}\left[\sum_{i \neq j \in [d]} \mu_{i,j}\phi_i(s,a)\phi_j(s,a)\right]$ is an orthogonality regularizer for a set of positive Lagrange multipliers $\{\lambda_{i,j}, \mu_{i,j}\}_{i \neq j \in [d]}$.

If $\phi^*, \tilde{\theta}^*$ is a solution to the primal problem in Equation (7), then there exist set of positive Lagrange multipliers $\{\lambda_{i,j}^*, \mu_{i,j}^*\}_{i \neq j \in [d]}$ such that $\phi^*, \tilde{\theta}^*$ satisfies the KKT conditions: (1) $\nabla\mathcal{L}(\phi^*, \theta^*; \nu, \rho) = 0$ and (2) $\Phi$ and $\tilde{\Theta}$ are orthogonal (Theorem 2.1 in Wright (2006)).

$\mathcal{L}_{orth}(\phi, \tilde{\theta})$ can be further relaxed into:
$$\mathcal{L}_{orth}(\phi, \tilde{\theta}) = \sum_{i \neq j \in [d]} \lambda_{i,j}(\mathbb{E}_{\nu(\theta)}\left[\tilde{\theta}_i(\theta)\tilde{\theta}_j(\theta)\right])^2 + \sum_{i \neq j \in [d]} \mu_{i,j}(\mathbb{E}_{\rho(s,a)}[\phi_i(s,a)\phi_j(s,a)])^2,$$
which can be simplified to the one proposed by Wu et al. (2018):

$$\mathcal{L}_{orth}(\phi, \tilde{\theta}) = \mathbb{E}_{\theta, \theta' \sim \nu(\theta)}\left[\sum_{i \neq j \in [d]} \tilde{\theta}_i(\theta)\tilde{\theta}_j(\theta)\tilde{\theta}_i(\theta')\tilde{\theta}_j(\theta')\right]$$
$$+ \mathbb{E}_{(s,a),(s',a') \sim \rho(s,a)}\left[\sum_{i \neq j \in [d]} \phi_i(s,a)\phi_j(s,a)\phi_i(s',a')\phi_j(s',a')\right].$$

## E.2   IMPLEMENTATION DETAILS

**Implementation of $\tilde{\theta}$ in SBM Loss.** The update for $\tilde{\theta}$ (implicitly or explicitly parameterized) within the spectral representation learning phase is crucial. One could approximate $\tilde{\theta}(\theta)$ by explicit parameters (reset each time new $\theta$ is sampled) or by a parameterized function (e.g. a neural network). In this work, we implement $\tilde{\theta}$ as a residual network: $\tilde{\theta}(\theta) = \theta + \Delta(\theta)$, where $\Delta(\theta)$ is a trainable MLP model.

**Moving Average of $\Lambda$.** Practically, SBM Loss is designed to be optimized in mini-batch iterative manner. Therefore, for better stability, we suggest using exponential moving average (EMA) updates: $\Lambda_{1,t+1} = \alpha\mathbb{E}_{\rho(s,a)}[\phi_t(s,a)\phi_t(s,a)^\top] + (1-\alpha)\Lambda_{1,t}$ (same for $\Lambda_2$) for some $\beta \in [0,1]$. This is a

solution for the constrained objective:

$$\Lambda_{1,t+1} \in \arg\min_{\Lambda} \frac{1}{2} \left\| \Lambda - \mathbb{E}_{\rho(s,a)}\left[ \phi_t(s,a)\phi_t(s,a)^\top \right] \right\|_2^2 + \frac{\eta}{2} \left\| \Lambda - \Lambda_{1,t} \right\|_2^2,$$

where $\eta > 0$ and $\alpha = \frac{1}{1+\eta}$.

**Network Architecture.** For the DQN architecture, we follow the architecture suggested by Mnih et al. (2013) for $\phi(s,a)$ where the last layer's output dimension was changed to $d \cdot |\mathcal{A}|$. The output is reshaped into a $|\mathcal{A}| \times d$ matrix such that for any input state $s$, we get $\phi(s,a)$ for each action $a \in \mathcal{A}$. The network in the residual model $\tilde{\theta}$ is a 3-layer MLP with dimensions of $2d \to 2d \to d$.

For R2D2 architecture, we follow the ResNet architecture from Espeholt et al. (2018) for the feature extraction from the visual observations $e_t = g(o_t)$. We use an LSTM (put a reference to LSTM here) network to process the sequence of observations. The LSTM head outputs a hidden state $m_t$ and a state $c_t$: $m_t, c_t = f(e_t, m_{t-1})$. The output feature $\phi(h,a)$ is done by using an MLP with output dimension of $|\mathcal{A}| \times d$, reshaped into a $|\mathcal{A}| \times d$ matrix such that for any input history $h_t = (o_t, o_{t-1}, \ldots o_0)$, we get $\phi(h_t, a)$ for each $a \in \mathcal{A}$. $\tilde{\theta}$ with the same architecture in the DQN case.

**Evaluation Parameters.** Each experiment ran for 10 different seeds. During evaluation, we ran the policy 10 times.

**Retrace Operator.** We used the transformed retrace operator $\mathcal{R}_f^\beta$ in our R2D2 experiments as target.

### E.3 HYPERPARAMETERS

Table 3: Hyperparameters for SBM with DQN

| | |
|---|---|
| $\gamma$ | 0.99 |
| Latent dimension $d$ | 256 |
| Learning rate | $3 \times 10^{-4}$ |
| Enviornment steps | 100M |
| Batch size | 256 |
| Number of representation learning steps | 512 |
| Number of policy optimization steps | 64 |
| Optimizer | Adam (Kingma, 2014) |
| $\sigma_{exp}$ | $\lambda_{min}(\Sigma)/d(1-\gamma)$ |
| $\sigma_{rep}^2$ | $10^{-2}$ |
| $\lambda$ | 0.1 |
| Reward clipping value | 1.0 |
| EMA parameter $\alpha$ | 0.1 |

Table 4: Hyperparameters for SBM with R2D2

| | |
|---|---|
| $\gamma$ | 0.997 |
| Latent dimension $d$ | 256 |
| Learning rate | $3 \times 10^{-4}$ |
| Enviornment steps | 100M |
| Batch size | 64 |
| Number of representation learning steps | 512 |
| Number of policy optimization steps | 64 |
| Number of policy optimization steps | 64 |
| Burn-in steps (Kapturowski et al., 2018) | 40 |
| Trajectory training (max) length (Kapturowski et al., 2018) | 120 |
| Retrace parameter $\beta$ | 0.95 |
| Optimizer | Adam (Kingma, 2014) |
| $\sigma_{exp}$ | $\lambda_{min}(\Sigma)/d(1-\gamma)$ |
| $\sigma^2_{rep}$ | $10^{-2}$ |
| $\lambda$ | 0.1 |
| EMA parameter $\alpha$ | 0.1 |

# F  ATARI SCORE TABLE

| Game | Human | Random | DQN | SBM+DQN | R2D2 | R2D2+SBM |
|---|---|---|---|---|---|---|
| Alien | 7127.70 | 227.80 | 1999.79 ± 257.86 | 1237.28 ± 108.98 | 6792.85 ± 295.58 | **8389.99 ± 384.19** |
| Amidar | 1719.50 | 5.80 | 433.66 ± 69.68 | 1159.18 ± 71.13 | 1575.45 ± 80.54 | **1596.91 ± 64.33** |
| Assault | 742.00 | 222.40 | 847.49 ± 54.91 | 1788.34 ± 61.37 | **2070.25 ± 285.82** | 2068.98 ± 362.52 |
| Asterix | 8503.30 | 210.00 | 6167.47 ± 1180.64 | **8681.20 ± 1182.53** | 6224.40 ± 615.35 | 7547.14 ± 438.34 |
| Asteroids | 47388.70 | 719.10 | 626.35 ± 74.15 | 1137.84 ± 69.64 | 1704.73 ± 193.96 | **2246.42 ± 159.39** |
| Atlantis | 29028.10 | 12850.00 | 497156.27 ± 4941.23 | 328266.40 ± 73334.77 | 903872.67 ± 97353.82 | **959381.33 ± 124777.47** |
| Bankheist | 753.10 | 14.20 | 320.85 ± 15.96 | **1136.08 ± 86.93** | 831.13 ± 34.91 | 983.09 ± 38.44 |
| Battlezone | 37187.50 | 2360.00 | 17066.67 ± 1426.73 | 26840.00 ± 1594.43 | 53993.33 ± 3267.35 | **65191.69 ± 3059.30** |
| Beamrider | 16926.50 | 363.90 | 2341.12 ± 384.89 | **5684.80 ± 411.47** | 4116.11 ± 404.47 | 5531.92 ± 299.30 |
| Berzerk | 2630.40 | 123.70 | 299.52 ± 36.49 | 746.67 ± 10.03 | **897.26 ± 92.56** | 841.47 ± 105.35 |
| Bowling | 160.70 | 23.10 | 21.72 ± 1.77 | 11.35 ± 2.58 | 228.65 ± 1.78 | **282.34 ± 1.29** |
| Boxing | 12.10 | 0.10 | 56.62 ± 1.04 | 95.24 ± 0.62 | 85.84 ± 1.17 | **100.00 ± 0.00** |
| Breakout | 30.50 | 1.70 | 74.07 ± 14.16 | 32.47 ± 2.82 | **197.29 ± 32.99** | 183.10 ± 23.84 |
| Centipede | 12017.00 | 2090.90 | 2429.27 ± 545.55 | 3164.83 ± 208.75 | 15324.34 ± 1810.83 | **16902.07 ± 1891.82** |
| Choppercommand | 7387.80 | 811.00 | 610.13 ± 91.47 | 915.20 ± 62.60 | 1753.27 ± 213.60 | **1950.43 ± 180.47** |
| Crazyclimber | 35829.40 | 10780.50 | 58781.87 ± 4431.60 | 91828.00 ± 7361.21 | 107756.13 ± 4010.05 | **136573.43 ± 3712.86** |
| Demonattack | 1971.00 | 152.10 | 4431.15 ± 562.47 | **8895.04 ± 712.73** | 2748.50 ± 397.09 | 2691.05 ± 468.96 |
| Doubledunk | -16.40 | -18.60 | -2.24 ± 1.45 | -2.53 ± 1.87 | 7.04 ± 2.21 | **8.77 ± 2.55** |
| Enduro | 860.50 | 0.00 | 638.51 ± 41.65 | **2313.70 ± 131.03** | 1811.51 ± 47.32 | 2305.42 ± 40.05 |
| Fishingderby | -38.70 | -91.70 | 3.20 ± 2.76 | -21.67 ± 4.62 | **35.73 ± 2.75** | 33.77 ± 2.69 |
| Freeway | 29.60 | 0.00 | 19.07 ± 0.21 | 31.29 ± 0.30 | 30.82 ± 0.07 | **33.00 ± 0.14** |
| Frostbite | 4334.70 | 65.20 | 2585.60 ± 278.19 | 535.04 ± 117.30 | 7726.51 ± 220.78 | **8734.95 ± 211.13** |
| Gopher | 2412.50 | 257.60 | 3546.45 ± 694.18 | **21192.16 ± 2595.47** | 9775.83 ± 2535.30 | 11532.03 ± 2724.36 |
| Gravitar | 3351.40 | 173.00 | 160.00 ± 49.53 | 242.00 ± 56.17 | **3370.03 ± 226.38** | 3316.33 ± 195.74 |
| Hero | 30826.40 | 1027.00 | 16431.15 ± 1016.03 | 21358.04 ± 1150.31 | 25955.32 ± 31.69 | **29876.39 ± 34.87** |
| Icehockey | 0.90 | -11.20 | -4.88 ± 0.69 | -5.43 ± 1.13 | 6.49 ± 1.71 | **6.80 ± 1.34** |
| Jamesbond | 302.80 | 29.00 | 445.87 ± 40.24 | 646.80 ± 40.03 | 721.93 ± 78.90 | **734.88 ± 88.68** |
| Kangaroo | 3035.00 | 52.00 | 2999.47 ± 333.14 | 7840.80 ± 680.31 | 11265.80 ± 478.40 | **12907.43 ± 561.41** |
| Krull | 2665.50 | 1598.00 | 4504.32 ± 207.00 | 11408.32 ± 248.90 | 21415.33 ± 3964.59 | **23311.90 ± 3284.52** |
| Kungfumaster | 22736.30 | 258.50 | 3136.00 ± 843.01 | 19676.80 ± 1179.52 | 34325.20 ± 1823.20 | **43780.17 ± 1773.79** |
| Montezumarevenge | 4753.30 | 0.00 | 0.00 ± 0.00 | 642.40 ± 113.34 | 879.67 ± 219.29 | **1176.24 ± 181.95** |
| Mspacman | 6951.60 | 307.30 | 2289.07 ± 296.70 | 2578.40 ± 186.92 | 6960.41 ± 353.38 | **8120.56 ± 378.52** |
| Namethisgame | 8049.00 | 2292.30 | **5047.89 ± 687.45** | 4769.60 ± 397.33 | 4038.58 ± 263.41 | 4894.12 ± 341.02 |
| Phoenix | 7242.60 | 761.40 | 4967.68 ± 753.66 | **6285.84 ± 153.60** | 4347.37 ± 355.99 | 3987.71 ± 444.14 |
| Pitfall | 6463.70 | -229.40 | -59.76 ± 10.22 | -14.03 ± 8.35 | -5.22 ± 2.51 | **-3.69 ± 3.23** |
| Pong | 14.60 | -20.70 | 10.28 ± 0.76 | 19.87 ± 0.22 | 18.62 ± 0.14 | **20.47 ± 0.18** |
| Privateeye | 69571.30 | 24.90 | -547.84 ± 84.07 | -181.95 ± 90.23 | 38212.54 ± 1274.77 | **51369.59 ± 1178.57** |
| Qbert | 13455.00 | 163.90 | 5128.53 ± 978.90 | 8241.20 ± 1016.85 | **20062.47 ± 835.18** | 18471.19 ± 703.67 |
| Riverraid | 17118.00 | 1338.50 | 6721.28 ± 510.13 | **12111.44 ± 712.36** | 7401.33 ± 308.99 | 9564.36 ± 282.00 |
| Roadrunner | 7845.00 | 11.50 | 33083.73 ± 1722.53 | **46015.20 ± 2269.45** | 42994.47 ± 2159.38 | 41576.14 ± 2726.37 |
| Robotank | 11.90 | 2.20 | 27.99 ± 5.36 | 17.86 ± 2.72 | 60.61 ± 1.81 | **62.75 ± 1.47** |
| Seaquest | 42054.70 | 68.40 | 2896.64 ± 363.70 | 2261.60 ± 209.35 | **3343.95 ± 195.17** | 3156.72 ± 234.17 |
| Skiing | -4336.90 | -17098.10 | -15384.64 ± 278.75 | **-15214.91 ± 1750.84** | -27300.00 ± 0.00 | -17024.68 ± 0.00 |
| Solaris | 12326.70 | 1236.30 | 1680.21 ± 421.35 | 3806.07 ± 44.73 | 3110.99 ± 378.79 | **4061.02 ± 491.34** |
| Spaceinvaders | 1668.70 | 148.00 | 1108.05 ± 267.36 | **1904.32 ± 138.75** | 1585.83 ± 198.53 | 1433.51 ± 226.71 |
| Stargunner | 10250.00 | 664.00 | **33361.07 ± 2218.71** | 6512.00 ± 849.11 | 2723.93 ± 548.09 | 2621.41 ± 690.72 |
| Tennis | -8.30 | -23.80 | 11.69 ± 0.47 | 13.31 ± 0.25 | 12.44 ± 2.66 | **14.26 ± 1.87** |
| Timepilot | 5229.20 | 3568.00 | 5230.93 ± 337.32 | 7594.40 ± 321.73 | 8032.27 ± 499.41 | **9550.81 ± 547.06** |
| Tutankham | 167.60 | 11.40 | 124.80 ± 7.70 | **189.64 ± 10.34** | 112.05 ± 18.01 | 134.66 ± 19.94 |
| Upndown | 11693.20 | 533.40 | 4451.84 ± 816.43 | 10883.84 ± 913.45 | 77144.95 ± 12594.93 | **99513.17 ± 9181.75** |
| Venture | 1187.50 | 0.00 | 439.47 ± 115.21 | 1182.97 ± 19.43 | 1365.00 ± 45.61 | **1949.20 ± 39.73** |
| Videopinball | 17667.90 | 0.00 | 97079.34 ± 22379.44 | **310351.10 ± 53324.81** | 16346.69 ± 2432.50 | 14943.88 ± 1760.91 |
| Wizardofwor | 4756.50 | 563.50 | 955.73 ± 453.05 | 2129.60 ± 136.21 | **10283.00 ± 1960.67** | 9889.49 ± 2038.26 |
| Yarsrevenge | 54576.90 | 3092.90 | 25562.24 ± 3743.63 | 27827.71 ± 2219.92 | 60559.04 ± 3913.49 | **73644.13 ± 3133.18** |
| Zaxxon | 9173.30 | 32.50 | 5619.20 ± 466.03 | 3238.40 ± 384.70 | 13134.33 ± 1342.95 | **17187.17 ± 1224.38** |

