# OpenReview forum: "Spectral Bellman Method: Unifying Representation and Exploration in RL"
_ICLR.cc/2026/Conference — ICLR 2026 Poster_

### Official Review · Reviewer_BNxx · 2025-10-16

**Soundness:** 2
**Presentation:** 3
**Contribution:** 3
**Rating:** 6
**Confidence:** 4

**Summary:**

This paper investigates representation learning in the low-inherent Bellman error (low-IBE) setting.

It introduces a new method and loss function, termed the Spectral Bellman Method (SBM).

The core insight lies in a fundamental spectral relationship. Under the zero-IBE condition, the Bellman operator’s transformation of the value function distribution is intrinsically tied to the feature covariance structure.

The framework can also be extended to the multi-step Bellman operator.

**Strengths:**

The paper is well-written, clearly structured, and easy to follow, making the technical content highly accessible.

The proposed SBM loss function is novel and represents a meaningful contribution to the literature.

Its motivation is well-justified, particularly since directly optimizing the MSE of IBE can be challenging and prone to inefficiencies.

The theoretical properties of the SBM loss and its connections to well-established techniques, such as power iteration, alternating optimization methods, and Thompson sampling, are especially compelling. These connections not only provide deeper insight into the behavior of the method but also situate it within a broader algorithmic context, highlighting potential avenues for further exploration.

The objective function itself appears more symmetric with respect to the parameters  theta and phi, which may contribute to improved stability and interpretability during optimization. Overall, the combination of novelty, theoretical grounding, and intuitive motivation makes the proposed approach both interesting and promising.

**Weaknesses:**

The paper appears to lack a formal theoretical guarantee for the minimizer of the SBM objective. For instance, it would be helpful to know whether there is any guarantee that the minimizer is close to the optimal value function, or more generally, how the SBM solution relates to the true underlying objective. Providing such a guarantee, or at least an informal discussion of its plausibility, would strengthen the theoretical contribution of the work.

One of the stated motivations of the paper is that the SBM objective is easier and more tractable to optimize than the traditional MSE objective. However, the paper does not provide clear evidence or empirical support for this claim. It would be valuable if the authors could elaborate on this point, for example by explaining why SBM optimization is expected to be more stable, faster, or less sensitive to hyperparameters, and by providing either empirical comparisons or theoretical intuition that substantiate the claim. This would help readers better understand the practical advantages of adopting the SBM formulation over conventional approaches.

**Questions:**

Is there any theoretical guarantee that the minimizer of the SBM objective is close to the optimal value function? Since the primary goal of reinforcement learning is typically to learn a near-optimal or optimal Q-function, it is crucial to understand whether the proposed loss function is well-aligned with this objective. Clarifying this connection would strengthen the theoretical foundation of the work and help readers assess the practical relevance of SBM.

On the practical side, could the authors comment on whether the SBM-based Q-learning method, particularly in its connection to Thompson Sampling, is easier or more efficient to implement in practice? Insights on implementation challenges, stability, or computational considerations would be valuable for understanding the real-world applicability of the approach.

It would also be helpful to discuss the computational complexity of the proposed algorithm, including how it scales with state-action dimensions, representation size, and sample complexity. A clear analysis would allow readers to compare the approach against existing methods and better understand its feasibility for larger or more complex problems.

Since the paper emphasizes representation learning, it would be useful to clarify whether the learned features provide meaningful benefits, both theoretically and empirically. Do the features improve policy learning, generalization, or sample efficiency?

Do the empirical experiments explicitly target the quality of these representations, or are the observed benefits mainly a byproduct of improved  Q-learning performance? Addressing these questions would help highlight the practical and theoretical significance of the representation learning component.

---

> ### Author Response · Authors · 2025-11-16
>
> We thank the reviewer for their positive feedback, finding our method to be a "meaningful contribution" with "compelling" theoretical properties. We appreciate the opportunity to clarify the theoretical guarantees and practical advantages of our method.
> 1. **Theoretical Guarantee & Alignment with $Q^*$ (Weakness 1 & Q1)**\
> That is a good point which we try to elevate throughout the paper. First, we establish that in the case of finding a representation with zero IBE (or even low IBE), an optimal solution can be easily found with efficient regret [1, 2], which is a significant benefit compared to the arbitrary neural network parametrized Q-learning.
> Theoretically, if the distributions of state-actions and weights were, for example, uniform over all possible states/weights, minimizing the SBM loss would guarantee zero/low IBE everywhere, leading to finding the optimal Q-function.
> However, our work mainly proposes a practical way to find such a representation in a fully online setting. To transfer this theoretical property into the applicative side of Deep RL, we make a few necessary relaxations to ensure feasibility: (1) We optimize the representation locally over the state-action distribution in our replay buffer and weights in the neighborhood of the current policy. (2) Instead of finding the minimizer in a batch setting, we perform mini-batch SGD with respect to the SBM loss.
> In our work, we cannot practically guarantee global coverage, so we aim for low IBE locally where our policy currently operates. By doing this, we leverage the desirable properties of low IBE in challenging real-world environments. Our experiments strengthen the claim that even aiming for low IBE locally provides significant performance gains and enables much more efficient exploration. Having said that, providing a formal theoretical guarantee for such local, online settings is left for future work.
>
> 2. **"Easier" Optimization & Empirical Evidence (Weakness 2 & Q2)**\
> We break down the stability and sensitivity aspects:\
> **Optimization Stability:** Direct MSE minimization (Eq 1) is often unstable because it relies on a noisy, single-sample estimate of the parameter covariance ($\hat{\Lambda} \approx \tilde{\theta}\tilde{\theta}^\top$). SBM decouples the problem into an alternating optimization (Algorithm 1) similar to Power Iteration. Crucially, SBM utilizes moving average covariance matrices from previous iterations. This batch-averaged influence provides a significantly more stable training signal than the high-variance gradients of the Naive MSE, preventing the representation from diverging or collapsing early in training.\
> **Hyperparameter Sensitivity:** SBM is primarily sensitive to the weight distribution std ($\sigma\_{rep}$) and dimension ($d$). Grid search identified $\sigma_{rep}^2 = 10^{-2}$ and $d=256$ as optimal. Increasing $d$ increased cost without performance gains.\
> **Empirical Results:** This stability enables SBM to consistently converge to effective representations, whereas the Naive MSE baseline often stagnates in suboptimal minima ($0.37$ vs $0.45$ HNS).
>
> 3. **Computational Complexity & Implementation (Q3)**\
> SBM introduces a  ~20% compute overhead per 1M steps over the vanilla versions, primarily due to training the $\tilde{\theta}$ network and TS covariance inversion. However, the representation learning cost remains comparable to baselines, and the Q-learning phase is highly efficient (linear updates only). We have added this runtime analysis to the revision."
>
> 4. **Isolating Representation Benefits (Q4 & Q5)**
> Definitely, our experiments explicitly target the representation quality by showing that the representation learned by SBM provides a superior basis for exploration. It achieves this by constructing a covariance structure $\Sigma$ that aligns with the Bellman operator. This benefit is clearly demonstrated in our comparisons against PVN and vanilla methods.\
> Furthermore, this benefit is clearly isolated in our updated ablation study (Table 2), where we hold the exploration algorithm (TS) constant across three different representations:\
> **New ablation:** Features learned via standard DQN loss (0.30 HNS on Explore),
> Features learned via Naive Bellman MSE (0.37 HNS), and
> Full SBM features (0.45 HNS).
> Since the exploration mechanism is identical in all three cases, the substantial performance gain, especially over the newly added standard DQN baseline, is entirely attributable to the SBM features providing a more accurate geometric "map" of uncertainty for the agent to follow.\
> We add the new ablation to the revised paper.
>
> [1] Zanette, et al. “Learning near optimal policies with low inherent bellman error.” In International Conference on Machine Learning 2020.\
> [2] Zanette, et al. “Provably efficient reward-agnostic navigation with linear value iteration.” 34th Conference on Neural Information Processing Systems (NeurIPS 2020)

---

> > ### Comment · Reviewer_BNxx · 2025-11-27
> >
> > Thanks for the response! I have read the response and other reviews. I will keep my score.

---

### Official Review · Reviewer_mbKF · 2025-10-28

**Soundness:** 2
**Presentation:** 2
**Contribution:** 2
**Rating:** 4
**Confidence:** 3

**Summary:**

The work attempts to combine representation learning and exploration for better performance in RL tasks. In particular, they focus on the Atari benchmark. The method attempts to learn zero IBE features and they combines it with Thompson sampling for better exploration. They combine their approach with two algorithm - DQN and R2D2 and show performance improvement, particularly on games which demand hard exploration.

**Strengths:**

* The work is well motivated, it makes sense to learn representations which directly support exploration/learning better strategies.
* The theory is well motivated and sound. Specifically, spectral decomposition under zero IBE conditions and the way to use power iteration for SBM loss optimisation are useful results.
* The performance gains are good when SBM + TS is used with baseline algorithms, this is in-particular substantial in the hard exploration games.
* The extension to multi-step operators (Retrace for R2D2) is theoretically sound and practical.

**Weaknesses:**

* Baseline: The paper is missing two critical baselines - DQN + TS or any DQN based method which uses TS (for instance BDQN (Azizzadenesheli et al. 2018)) but does not use spectral features. Similar is the case with R2D2, the work is missing a critical baseline R2D2 + TS without spectral features (which I understand is might be non-trivial to implement but without this, it is very hard to truly guage SBM's effectiveness). This is extremely important to have to be able to gauge the efficacy of SBM features. The paper cites Azizzadenesheli et al. (2018) in their related work section but provides no justification for excluding it as a baseline. The current experimental design confounds two variables (representation learning + exploration strategy), making it difficult to attribute performance gains to either component independently. For instance, from table 1, it seems the major chunk of improvement is coming after adding TS to SBM (its good enough for DQN but with R2D2, R2D2 + eps → R2D2 + SBM + eps the improvement is pretty small (basically none in terms of median scores), whereas R2D2 + SBM + eps → R2D2 + SBM + TS shows +0.23 improvement overall and 0.16 on explore subset). This pattern suggests TS may contribute more to the gains than the spectral representation itself, which contradicts the paper's central claim that SBM representations "naturally facilitate structured exploration."
* For exploitation-heavy, precision-control games (breakout, frostbite), the performance sees a decline from the baseline algorithm (DQN specifically) which I am guessing is due to the fact that Thompson sampling may add harmful exploration noise. Authors should discuss a bit about this and suggest if there are ways to control this (e.g., annealing the Thompson Sampling variance parameter or adaptive mechanisms to reduce exploration noise when sufficient policy convergence is achieved). Some ablations and experiments showing when SBM helps vs. hurts performance would strengthen the paper. However, this is a minor thing.
* The Thompson Sampling implementation (Equation 5) directly follows Zanette et al. (2020b) without modification. The only claimed contribution is using SBM-learned features in the covariance, but without DQN+TS baselines, it's unclear if standard representations would work equally well.
* The paper provides no analysis of computational overhead with baselines.

**Questions:**

1. Can you provide results for DQN + TS and R2D2 + TS without spectral features? If not, any justification as to why there were excluded.
2. In Table 1, R2D2 → R2D2+SBM shows almost no improvement (in terms of median score for the explore subset), while R2D2+SBM → R2D2+SBM+TS shows +0.16 improvement. Doesn't this suggest TS contributes more than SBM? How do you reconcile this with your claim that SBM representations "naturally facilitate structured exploration"?
3. What is the computational overhead of alternating between Q-learning and SBM representation learning?
4. On games like Breakout and Frostbite, SBM+TS performs worse than baseline DQN. Can you discuss this? Is this due to harmful exploration noise in exploitation-heavy games? Can you discuss/analyze this? However, as stated, this is a minor requirement.

---

> ### Author Response · Authors · 2025-11-16
>
> We thank the reviewer for their detailed critique. We appreciate that you found the work "well motivated". We value your rigorous check on baselines and believe clarifying the experimental design plus additional ablation experiment we did will address your main concerns.
> 1. **Missing Baselines without spectral features (Weakness 1/3 & Q1)**\
> You are correct that isolating the effect of the representation is critical. We address this through two distinct baselines in Table 2:\
> **Existing Baseline ("Features from Naive MSE Loss"):** This baseline, present in the original submission, uses a DQN agent optimizing the naive IBE minimization (Eq 1) over a distribution of parameters. Even with this broader distribution, this baseline achieves inferior results compared to SBM + TS.\
> **New Baseline ("Features from DQN Loss"):** To fully align with your request for a standard "DQN + TS" baseline, we conducted a new ablation where the parameter distribution is collapsed to a Dirac delta ($\nu_t(\theta) = \delta(\theta - \hat{\theta}_t)$). This effectively restricts representation learning to the current policy only (standard DQN loss) while attempting to use the resulting feature covariance for TS. This configuration performed even worse, achieving an HNS of 0.30.\
> In conclusion, both baselines significantly underperform SBM + TS (0.45). This strongly supports our central claim: TS is not a "plug-and-play" module. It relies on the specific geometry of the feature space to estimate uncertainty. Standard DQN features, whether optimized for a single policy (DQN Loss) or a distribution (Naive MSE), fail to capture the spectral structure of the Bellman operator, rendering their covariance a poor proxy for true epistemic uncertainty.\
> We chose DQN because it provides a simpler, cleaner setting to isolate the representation's impact. The advantage of spectrally aligned covariance is a fundamental geometric property, independent of the specific RL backbone (DQN vs. R2D2). Validating this on DQN is sufficient to prove the core hypothesis without the added complexity of R2D2.\
> We thank the reviewers for their helpful comment, and we updated the paper with the new ablation results.
> 2. **Disentangling SBM vs. TS (Weakness 1 & Q2)**\
> You noted that R2D2 $\to$ R2D2+SBM yields small gains, while adding TS yields large gains, suggesting "TS contributes more."
> We view this differently: TS exploits the structure that SBM creates. SBM learns a representation where the covariance matrix $\Sigma$ reflects the Bellman operator's structure (the "map"). Epsilon-greedy strategies (R2D2+SBM) ignore this map, resulting in only minor gains from better value approximation. TS (R2D2+SBM+TS) actively uses this map to drive exploration.
> The large jump in performance when adding TS to SBM confirms that SBM successfully learned a structured covariance. If SBM had failed to learn meaningful features, adding TS would have simply added noise and likely hurt performance (as seen in the Naive MSE ablation in Table 2). Thus, the gains are not by only using TS,  but the synergy of TS acting with SBM features.
> 3. **Performance Drops in Precision Games (Weakness 2 & Q4)**\
> That’s a great observation that we actually missed!\
> We agree that the performance drop in precision games like Breakout stems from optimistic exploration in instant-death environments, where high-variance actions often terminate episodes. This probably can be mitigated by annealing the exploration noise or switching to greedy evaluation once a threshold is reached. However, in order to keep things as simple as possible, we kept hyperparameters constant across all games to avoid tuning per-environment.\
> To confirm this is an exploration tuning issue rather than a representation flaw, we ran a follow-up experiment on Breakout using linear noise annealing (decaying to $10^{-5}$ over 20M steps). This fully recovered performance to $92.38 \pm 5.31$ (DQN+SBM) and $221.00 \pm 17.51$ (R2D2+SBM). We have added a discussion of this trade-off and these results to the revised paper.
> 4. **Computational Overhead (Weakness 4 & Q3)**\
> The computational overhead in SBM is relatively small. It primarily stems from the additional training of the network for $\tilde{\theta}$ in our implementation. The representation learning process itself has a cost similar to vanilla baselines. Notably, the Q-learning phase is highly efficient because we only train the linear weights of the current policy with respect to the learned representation from the representation phase. Additional overhead comes from the TS itself, where before each rollout the inverse of the covariance needs to be computed. Per 1M environment steps we measured a ~20% increase in compute time. We add this runtime clarification to the revised version

---

> ### Comment · Reviewer_mbKF · 2025-11-16
> **Response to the authors**
>
> I thank the authors for their response. In the context of the new results posted by the authors + their hypothesis about the synergy between SBM features and TS (which I agree with), I believe most of my doubts which were preventing me from recommending an acceptance are cleared. I have increased my score to reflect this.
>
> However, I still have one question - isn't Azizzadenesheli et al. (2018) also a potential baseline? This is exactly the DQN type suitable for TS. While now I understand the contribution of your work in a much better fashion, I am still not sure if the baseline coverage is full, Azizzadenesheli et al. (2018) seems to be performing really well relative to DQN and also use TS, therefore your contribution will become even stronger if positioned with this method as well. Rather than Naive DQN + TS, I believe Azizzadenesheli et al. is a much closer comparison to SBM + DQN + TS.
>
> Ref:
>
> [1]Efficient Exploration through Bayesian Deep Q-Networks. Kamyar Azizzadenesheli, Animashree Anandkumar., 2018.

---

> ### Author Response · Authors · 2025-11-17
>
> We sincerely thank the reviewer for raising their score. We are glad that the additional results have cleared your doubts.\
> Regarding your question on Azizzadenesheli et al: You are absolutely correct that this is a relevant baseline for establishing the value of our method. We are happy to clarify that the "DQN + TS" baseline we have just added in our additional results (and referred to as "Features from DQN Loss (DQN + TS) " in the ablation study) corresponds **exactly** to the method proposed by Azizzadenesheli et al. We will make this clarification in the paper, referring to the work of  Azizzadenesheli et al.

---

> > ### Comment · Reviewer_mbKF · 2025-11-17
> > **Response to authors**
> >
> > Thank you for your clarification and apologies for overlooking that. I am satisfied with all responses and now confident of my score. All the best!

---

### Official Review · Reviewer_pqsD · 2025-10-30

**Soundness:** 3
**Presentation:** 3
**Contribution:** 3
**Rating:** 6
**Confidence:** 2

**Summary:**

This paper introduces the Spectral Bellman Method (SBM), a novel framework designed to unify representation learning and efficient exploration in reinforcement learning.The central problem addressed is that these two critical components are often treated as separate modules, failing to leverage their potential synergy. The motivation stems from the insight that while representations with low Inherent Bellman Error (IBE) are highly desirable for value-based methods, directly optimizing for them is computationally intractable. The authors identify a key spectral relationship: under the ideal zero-IBE condition, the structure of the Bellman operator is fundamentally linked to the covariance structure of the features themselves. To solve this, SBM proposes a new learning objective inspired by the power iteration method. Instead of directly minimizing Bellman error, this objective encourages the feature covariance to align with the Bellman dynamics across a distribution of value functions. This serves as a tractable proxy for learning low-IBE representations. The method is integrated into an alternating training loop where the learned feature covariance is naturally used to drive exploration via Thompson Sampling, which in turn collects data to improve the policy and refine the representation learning target.

**Strengths:**

Strength:
1.	The paper introduces a novel objective for representation learning by reframing the intractable problem of minimizing Inherent Bellman Error (IBE) into a more tractable proxy based on the spectral properties of the Bellman operator.

2.	The framework provides a tight, natural coupling between representation and exploration. The feature covariance matrix learned by SBM is directly used to guide Thompson Sampling, creating a coherent feedback loop where better representations inform more structured exploration.

**Weaknesses:**

Major：
1.	The theoretical motivation for the SBM objective, particularly the spectral decomposition outlined in Theorem 1, is quite elegant. This derivation hinges on the ideal assumption of zero Inherent Bellman Error (IBE), where the function space is perfectly closed under the Bellman operator. I am curious about how the proposed framework is expected to behave when this assumption is inevitably relaxed in practice, as is the case when using complex neural network approximators. Could the authors provide some additional intuition or analysis on the robustness of the SBM objective when the IBE is small but non-zero? For example, how does the connection between the feature covariance and the Bellman operator change in this scenario? Does the SBM loss still serve as a more effective proxy for minimizing IBE than the standard MSE objective? Adding a brief discussion on this point would be very helpful in bridging the compelling theory with the practical algorithm.

2.	The paper's alternating optimization between policy and representation is interesting. However, I have a conceptual question about the representation learning target. Since the SBM objective is tied to the Bellman operator defined by the current policy, it seems the representation might become overly specialized to a suboptimal policy early in training. Could this create a "representational trap" that makes it difficult for the agent to later discover and represent a truly optimal policy? I would appreciate the authors' perspective on this potential issue.

**Questions:**

1. Could the authors elaborate on the behavior of the SBM objective when the Inherent Bellman Error (IBE) is small but non-zero? Specifically, how does the relationship between the feature covariance and the Bellman operator evolve in this case, and does the SBM loss remain a more faithful proxy for minimizing IBE than the standard MSE objective?

2. Given that the SBM objective depends on the Bellman operator of the current policy, how do the authors prevent the learned representation from becoming overly specialized to a suboptimal early policy? Could such coupling lead to a “representational trap,” and if so, what mechanisms or design choices mitigate this risk?

---

> ### Author Response · Authors · 2025-11-16
>
> We thank the reviewer for the positive feedback by accurately summarizing our work as a "novel objective" with a "tight, natural coupling between representation and exploration." We particularly appreciate your insightful questions regarding the robustness of the spectral condition and the dynamics of the representation learning.
>
> 1. **Robustness to Non-Zero IBE (Weakness 1 & Q1)**\
> We agree that the strict zero-IBE condition is unlikely to hold in practice for complex problems with function approximation. Our work is motivated by the structural properties derived under the zero-IBE assumption (Theorem 1), which guides the design of the SBM loss. However, zero-IBE is not the assumption required for SBM to function, but rather its optimization goal. As established by Zanette et al. [1] , the regret of value-based RL algorithms scales with the magnitude of the Inherent Bellman Error (IBE), specifically appearing as a $\tilde{O}(\dots + \sqrt{d}\mathcal{I}K)$ term in the regret bound. Their analysis demonstrates that performance degrades gracefully in the presence of non-zero IBE. Therefore, achieving low IBE, rather than strictly zero, is theoretically sufficient for effective learning, and it’s backed up by our experiments.\
> SBM explicitly aims to minimize this error. To make this tractable in deep RL, our SBM loss is minimized over a specific state-action distribution from the replay buffer and a parameter distribution, sampled locally around the current policy parameters. This effectively concentrates the representation learning capacity on reducing Bellman inconsistency in the relevant parts of the state-action space and for value functions close to the current policy. Our experiments show that optimizing for this spectral objective yields a lower effective IBE for the task at hand.\
> Moreover, we show in the paper that SBM provides statistically grounded regularization using moving average covariance matrices, whereas the standard MSE uses noisy, single-sample estimates, making SBM more robust even when the zero-IBE condition is violated.
> 2. **The "Representational Trap" & Suboptimal Policies (Weakness 2 & Q2)**\
> That is a great question. First, it is important to note that the Optimal Bellman Operator used in our framework is not policy-dependent. It is defined by the environment dynamics and the maximization over actions, and is therefore not defined by the current policy.\
> Nevertheless, you are correct that the state-action distribution and weight distribution in the SBM objective are indeed correlated with the current policy due to practical implementation reasons. The SBM loss specifically aims to find a linear representation with respect to these distributions.\
> As in any Deep RL framework, there is always a risk of converging to a suboptimal solution. However, the SBM framework is arguably far more robust to such "traps" for two reasons:\
> **Exploration Feedback Loop:** The Thompson Sampling (TS) mechanism explicitly encourages the policy to reach unseen states with high uncertainty , constantly pushing the distributions and to expand.\
> **Distributional Robustness:** Enforcing the representation to be linear with respect to a distribution of weights rather than a single point estimate as in vanilla Q-learning acts as a powerful regularizer. This requirement prevents the features from overfitting to a single value estimate, robustifying the method against collapsing into a degenerate representation.
>
> **[1] Zanette, et al. “Learning near optimal policies with low inherent bellman error.” In International Conference on Machine Learning 2020.**

---

### Official Review · Reviewer_5nzo · 2025-11-03

**Soundness:** 4
**Presentation:** 3
**Contribution:** 3
**Rating:** 8
**Confidence:** 3

**Summary:**

The paper proposes a new approach, the Spectral Bellman Method, for learning representations that aid exploration. The approach is built upon the Inherent Bellman Error (IBE) condition, leveraging the key insight that connects feature covariance and zero-IBE to derive an algorithm that alternates between learning features and Q-values. Furthermore, the method also leverages the underlying structural properties to enhance exploration via the Thompson sampling technique. Experiments conducted on the full Atari game suite show improvements over other baselines, observing significant gains on deep exploration games and corroborating the effectiveness of the approach.

**Strengths:**

- The paper presents a well-executed idea. The proposed method has solid mathematical backing, and the derived algorithm stems directly from the theoretical insights. It is encouraging to see these insights translate into meaningful empirical gains.
- The key insight regarding the spectral properties and the zero-IBE condition is particularly elegant, as it helps reduce a complex optimization problem to a more tractable one. It is a further strength that the Thompson sampling exploration fits well with the approach, tackling both feature learning and exploration simultaneously.
- The resulting algorithm represents only a minor modification to existing algorithms, such as DQN or R2D2, yet it achieves substantial empirical gains on many Atari games, including those where exploration is challenging.

**Weaknesses:**

- The mathematical derivations in Section 3 could be significantly improved for clarity. As written, the section is overly dense. The authors should consider simplifying complex notations (e.g., $\tilde{\theta}(\theta)$) and adding intuitive explanations to make the theoretical statements more comprehensible
- The paper would also benefit from a small-scale, illustrative experiment. Such an experiment would be valuable for building intuition and helping isolate the source of the method's benefits (representations, exploration, or both). Additionally, a discussion or analysis of the method's sensitivity to key parameters (such as the choice of state-action distribution pairs and Q-function parameters) is a notable omission.
- The experimental comparison to other representation learning methods is narrow, as it only includes one such baseline (Online-PVNs). A more thorough evaluation against other common baselines&mdash;particularly those using an auxiliary loss for representation learning&mdash;is needed to fully contextualize the proposed method's performance. This is especially true since several such methods are already discussed in the related work section.

**Questions:**

- The paper's claim that "effective exploration aims to reduce uncertainty" (lines 110-112) is presented as a general-purpose objective for exploration. The authors should justify this framing, as it seems to imply that exploration is also a problem in policy evaluation, not just control.
- The paper's assertion (line 149) that not resulting in task-agnostic representations is a feature is counterintuitive. Given that task-agnostic representations are generally desirable for fast adaptation to changing goals, the authors should provide a clear justification for why their task-primary approach is a benefit and not a limitation.
- The introduction of Equation 2 lacks sufficient motivation and derivation. It appears abruptly in the text, and the authors should provide more insight into how this equation was derived and why it is the correct formulation in this context.
- Building on the weakness of the missing ablation studies, the paper should provide the intuition or justification for the specific choices of parameters used in the experiments (e.g., for the state-action distribution and Q-function). An explanation of why these particular choices were made is necessary.
- The algorithm, as presented in the pseudocode, appears to operate on a batch of data for each update. This raises questions about the method's feasibility and efficiency in a fully online or streaming setting. The authors should clarify whether their approach can be adapted for such scenarios.

---

> ### Author Response · Authors · 2025-11-16
>
> We thank the reviewer for the positive assessment and for highlighting the "elegance" of our spectral insight and the "substantial empirical gains" achieved by our method. We appreciate the constructive feedback regarding the presentation and baselines. We address your specific questions below.
> 1. **Clarity of Mathematical Derivations & Equation 2 (Weakness 1 & Q3):** We acknowledge that Section 3 is dense. In the revised version, we simplify the notation (specifically $\tilde{\theta}(\theta)$) and provide more intuitive explanations alongside the formal statements. Regarding Equation 2, it is a crucial normalization step. In standard SVD, the decomposition applies to a raw matrix. However, our objective involves expectations over distributions $\rho(s,a)$ and $\nu(\theta)$. Equation 2 essentially "absorbs" the square roots of these probabilities into the features and parameters (e.g., $\bar{\phi} = \sqrt{\rho}\phi$). This allows us to treat the weighted problem as a standard spectral decomposition on the augmented matrices $\Phi_P$ and $\tilde{\Theta}_P$. We will add a dedicated remark clarifying this derivation.
> 2. **Intuition on Distributions and Sensitivity (Weakness 2 & Q4)**
> You asked for intuition regarding our distribution choices:\
> **State-Action distribution ($\rho$):** We use the replay buffer distribution. This is standard for off-policy RL and ensures the representation minimizes error on states the agent actually visits.\
> **Parameter distribution ($\nu$):** We define $\nu(\theta)$ as a Gaussian centered on the current policy $\hat{\theta}\_t$. The intuition is locality: we focus representation capacity on value functions relevant to the current stage of learning, rather than the entire parameter space.\
> **Parameter sensitivity:** We observed that SBM is sensitive to the distribution width $\sigma\_{rep}$. A grid search identified $\sigma\_{rep} = 10^{-2}$ as the optimal configuration.\
> In the revised version, we make these explanations clearer and with more details.
> 3. **Comparisons to Baselines (Weakness 3)**: We focused on Online PVN because it is the most relevant state-of-the-art baseline that also uses spectral/Laplacian properties for representation.Many auxiliary tasks (like contrastive learning) focus on visual features rather than the Bellman structure. These can be used as Orthogonal auxiliary losses to SBM. \
> Please note that Table 2  serves as a crucial baseline. We compared SBM against "Features from Naive MSE Loss" (Equation 1). The significant performance gap confirms that the improvement comes specifically from the spectral constraint, not just from having an auxiliary loss. In addition, we add a **new baseline** where the parameter distribution is a Dirac delta ($\nu_t(\theta) = \delta(\theta - \hat{\theta}_t)$). This effectively restricts representation learning to the current policy only (standard DQN loss) while attempting to use the resulting feature covariance for TS. This baseline yield even worse results, showing the benefit of using our proposed spectral representation learning method.  We add the new baseline to the revised paper.
> 4. **Task-Agnostic vs. Task-Specific Representations (Q2):**
> Thanks for bringing this interesting topic. We argue this is a trade-off between task-agnostic vs. task-specific.
> Task-Agnostic (e.g., PVN/SF): Excellent for transfer across rewards but may underfit the specific geometry of the current difficult task.
> Task-Specific (SBM): By enforcing closure under the optimal Bellman operator (which depends on the specific task reward function), SBM learns features optimized to approximate the value function of the task at hand. Our results suggest this specificity is beneficial for maximizing scores in hard-exploration environments.
> 5. **Exploration Framing (Q1):** We state that "effective exploration aims to reduce uncertainty" to reflect the objective of minimizing the Maximum Uncertainty (Definition 3) across all policies. As discussed in Section 2.2, reducing this global uncertainty requires finding a policy that covers the state-action space effectively under the given representation. This is achieved by estimating the value function (policy evaluation) optimistically in unvisited or high-uncertainty regions (via TS). Consequently, accurate policy evaluation, specifically quantifying uncertainty, is the mechanism that drives the control policy to explore and cover the state space
> 6. **Batch vs. Online Feasibility (Q5):** You are correct to point out that the Algorithm 2 pseudocode indicates batch-learning. In practice, as in many similar cases, we approximate the minimizers using mini-batches sampled from the replay buffer. We do not require full-batch SVD. Instead, we minimize the SBM Loss (which performs implicit power iteration steps) using stochastic gradient descent on these mini-batches. This is also true for the Q-learning phase. Therefore, SBM was actually tested in a fully

---

> > ### Comment · Reviewer_5nzo · 2025-11-24
> >
> > I thank the authors for their follow-up response. Although the final sentence appears to be cut off, the provided explanation sufficiently clarifies my questions; I will maintain my current evaluation.

---

> > > ### Author Response · Authors · 2025-11-24
> > >
> > > We thank the reviewer for the follow-up and are glad that our explanation addressed your concerns. We apologize that the final sentence of our previous message was cut off. We provide the full response regarding Batch vs. Online Feasibility below:
> > >
> > > 6. **Batch vs. Online Feasibility (Q5):**  You are correct to point out that the Algorithm 2 pseudocode indicates batch-learning. In practice, as in many similar cases, we approximate the minimizers using mini-batches sampled from the replay buffer. We do not require full-batch SVD. Instead, we minimize the SBM Loss (which performs implicit power iteration steps) using stochastic gradient descent on these mini-batches. This is also true for the Q-learning phase. Therefore, SBM was actually tested in a fully online setting (similar to DQN or R2D2). We have added this important clarification to the revised version. Thank you.

---

### Meta-Review · Area_Chair_eFRZ · 2026-01-05

**Summary:**

This submission proposes the Spectral Bellman Method (SBM), a representation learning objective motivated by spectral structure under the zero Inherent Bellman Error (IBE) condition, and integrates it with Thompson Sampling (TS) to enable structured exploration. Across reviewers, there is broad agreement that the core idea is novel and elegant, and that the approach yields strong empirical performance, especially on hard-exploration Atari games. Overall, the rebuttal substantially strengthened the paper by clarifying theoretical framing, adding missing ablations/baselines (esp. TS with non-SBM representations), and addressing practicality concerns. Most remaining concerns are now about presentation depth and breadth of comparisons, rather than fundamental correctness. Most concerns are addressed, and there will be a consensus of acceptance, thus acceptance is recommended.

**Reviewer Concerns:**

Most of the major concerns have been addressed. Some outstanding ones or could be improved.

Reviewer 5nzo flagged that Section 3 is dense and would benefit from clearer derivations and intuitive examples.
Reviewer notes comparison against representation learning baselines is narrow (only Online-PVNs).

**Reviewer Scores:**

Reviewer 5nzo (Rating: 8, confident accept)
Reviewer pqsD (Rating: 6, marginal accept)
Reviewer mbKF (Rating: 4 initially, then explicitly said to increase the score
Reviewer BNxx (Rating: 6, stayed at 6)

So after the rebuttal, there is a consensus of acceptance. AC agreed with the consensus and recommended acceptance.

---

### Decision · Program_Chairs · 2026-01-26

Accept (Poster)